# Synthetic *BZLF1*-targeted transcriptional activator for efficient lytic induction therapy against EBV-associated epithelial cancers

Man Wu [1,2,19], Pok Man Hau[1,19], Linxian Li[3,4,5,19], Chi Man Tsang [1,2], Yike Yang [3,6], Aziz Taghbalout[7], Grace Tin-Yun Chung[1], Shin Yee Hui [1], Wing Chung Tang[1], Nathaniel Jillette[7], Jacqueline Jufen Zhu [7,8], Horace Hok Yeung Lee[3], Ee Ling Kong[1], Melissa Sue Ann Chan[1], Jason Ying Kuen Chan [9], Brigette Buig Yue Ma [10], Mei-Ru Chen[11], Charles Lee [7], Ka Fai To [1,2], Albert Wu Cheng [7,8,12,13,14,15,16,17,18,20] ✉ & Kwok-Wai Lo [1,2,20] ✉

The unique virus-cell interaction in Epstein-Barr virus (EBV)-associated malignancies implies targeting the viral latent-lytic switch is a promising therapeutic strategy. However, the lack of specific and efficient therapeutic agents to induce lytic cycle in these cancers is a major challenge facing clinical implementation. We develop a synthetic transcriptional activator that specifically activates endogenous *BZLF1* and efficiently induces lytic reactivation in EBV-positive cancer cells. A lipid nanoparticle encapsulating nucleoside-modified mRNA which encodes a *BZLF1*-specific transcriptional activator (mTZ3-LNP) is synthesized for EBV-targeted therapy. Compared with conventional chemical inducers, mTZ3-LNP more efficiently activates EBV lytic gene expression in EBV-associated epithelial cancers. Here we show the potency and safety of treatment with mTZ3-LNP to suppress tumor growth in EBV-positive cancer models. The combination of mTZ3-LNP and ganciclovir yields highly selective cytotoxic effects of mRNA-based lytic induction therapy against EBV-positive tumor cells, indicating the potential of mRNA nanomedicine in the treatment of EBV-associated epithelial cancers.

Epstein-Barr virus (EBV) is the first cancer-associated virus identified in humans and it affects more than 90% of the global population. While EBV carriers remain mostly asymptomatic lifelong, latent EBV infection contributes to the transformation and progression of a variety of human malignancies, including endemic Burkitt lymphoma (BL), Hodgkin's lymphoma, natural killer (NK)-/T-cell lymphoma, post-transplant lymphoma, nasopharyngeal carcinoma (NPC) and a subset of gastric cancers (GC). These cancers have a global incidence rate of more than 265,000 persons/year and a mortality rate of more than 164,000 persons/year[1,2]. All EBV-associated tumors share unique features including the continued presence of multiple viral genomes (up to 100 genomes/cell) and a restricted latency program. In these malignancies, EBV maintains episomal genomes and expresses multiple latent genes to modulate cancer hallmarks. The presence of viral episomal genomes in all EBV-associated cancer cells serves as a tumor-specific target for the development of efficient therapeutic strategies against these cancers[2,3].

Switching the EBV-infected cells from latent to lytic cycle can induce growth arrest, promote apoptosis, and cause lytic rupture of the cells, and thereby being an attractive approach to curing EBV-associated malignancies[4–9]. When a latent EBV is induced into the lytic cycle, the immediate-early (IE) proteins, Zta and Rta encoded by

---

*BZLF1* and *BRLF1*, respectively, must be expressed. These proteins then activate the transcription of a panel of early (e.g., EA-D, BGLF4) and late (e.g., VCA, gp350) proteins to facilitate the lytic replication of EBV genomes and production of infective virions[7–9]. Reactivation of EBV from latency is dependent on the expression of the viral Zta protein. Ectopic *BZLF1* expression alone can trigger the switch from the latent to the lytic stage and drive EBV lytic cycle completion in EBV-infected cells. Therefore, cytolytic virus activation (CLVA), in which chemical lytic inducers (e.g., gemcitabine, valproic acid, sodium butyrate, and other histone deacetylase inhibitors) are used to trigger EBV to enter the lytic phase, has been developed as a therapeutic strategy to specifically target EBV-associated cancers. In this type of lytic induction therapy, antiviral ganciclovir (GCV) is co-administered with the chemical inducer to patients to mediate specific cell killing and prevent virus production in EBV-infected cells. GCV is non-cytotoxic to EBV-positive tumors with restricted viral latency. However, BGLF4, an EBV-encoded serine/threonine kinase is expressed during lytic reactivation and can convert the non-cytotoxic GCV to an active, cytotoxic form via phosphorylation. This BGLF4-converted cytotoxic GCV or phosphorylated GCV rapidly kills cancer cells and has been evaluated in clinical trials of oncolytic therapy against EBV-associated cancers[3,7–9]. In addition to mediating the direct killing of EBV-positive tumor cells after lytic reactivation, the phosphorylated GCV can be transferred to adjacent cells, leading to a "bystander killing" effect. Importantly, phosphorylated GCV can inhibit EBV-encoded DNA polymerase, interrupting the production of infective virions and preventing dissemination of virus during lytic induction therapy[7–11].

CLVA treatment has been evaluated in phase-I/II clinical trials involving patients with recurrent NPC and has elicited a clinical response in some patients[10,11]. However, the efficiency of chemical activators for inducing lytic reactivation in EBV-associated tumors often has been low and variable. These activators also have low specificity for EBV activation and a broad spectrum of cytotoxicity, meaning that the treated cells probably die from the toxic effect of the chemicals before viral lytic reactivation is induced. Recent studies have revealed that chemical activator-induced EBV lytic reactivation is cell context-specific. The efficiency of chemical activator-based lytic induction therapy depends upon a variety of acquired epigenetic changes and cellular transcription factors in the tumor cells[7–9]. In native EBV-associated gastric cancer (EBVaGC) and NPC cell lines (e.g., SNU719, C666-1, NPC43, and C17), weak or no lytic gene expression was detected after treatment with various chemical inducers (Supplementary Fig. 1). Only small proportions of tumor cells with lytic gene expression was observed, even in tumors that responded to treatment with chemical inducers. None of the reported chemical inducers could universally reactivate the lytic cycle in all native EBV-positive epithelial cancer cell lines.

In this work, we explore whether a synthetic *BZLF1*-speciific transcriptional activator could bypass the restrictions imposed by various cellular factors on EBV reactivation and improve the specificity of lytic induction therapy. By exploiting the CRISPR-Casilio activator system, we demonstrate the feasibility of using an artificial activator to reactivate EBV lytic genes and the cytotoxic effect of this activator in NPC and EBVaGC cells[12]. The complexity of the system potentially poses delivery challenges that reduce the efficiency of in vivo activation of *BZLF1* in EBV-positive epithelial cancers and limits its clinical applications. In contrast to the CRISPR-Casilio system, the simplicity of the transcription activator-like effector (TALE)-based transcriptional activator system implies its usefulness in the development of EBV lytic induction therapy for clinical applications[13]. To achieve successful in vivo therapeutic delivery of *BZLF1*-specific artificial transcriptional activation system, nucleoside-modified mRNAs encoding this *BZLF1*-specific TALE-transcriptional activator are synthesized and encapsulated in formulated lipid nanoparticles (LNPs) for efficient delivery to EBV-associated epithelial cancers and induction of the EBV immediate early lytic gene *BZLF1* transcription, subsequently switching toward the lytic cycle in tumor cells. The LNP-encapsulated mRNA encoding a *BZLF1*-specific TALE-transcriptional activator induces the EBV lytic cycle with high efficiency and is universally applicable to all EBV-positive epithelial cancers. The potent and specific cytotoxic effect of this EBV-targeted mRNA drug in both in vitro and in vivo tumor models implies its potential as a promising nanomedicine therapy for EBV-associated malignancies.

## Results

### CRISPR-Casilio activator system induces EBV lytic reactivation

To reactivate the EBV lytic cycle in EBVaGC and NPC cells, we first exploited the efficient CRISPR-based Casilio activator system to induce the endogenous expression of the EBV-encoded immediate early lytic gene *BZLF1*. The Casilio activator system consists of a dCas9 protein, a PUFa-p65HSF1 activator module, and a sgRNA appended with five copies of PBSa[12]. It was specifically designed for recruitment of multiple PUF-p65HSF1 activators to enable the potent transcriptional activation of target genes. Given the high feasibility of sgRNA synthesis, this system can be used to screen multiple target sequences for efficient transcriptional activation. Here, the EBVaGC cell line SNU719 and the NPC cell line C666-1 were co-transduced with lentiviral vectors encoding HA-dCas9-EGFP, 3xFLAG-PUFa-p65HSF and a panel of sgRNAs-5xPBSa that can bind to the *BZLF1* promoter (sgRNA1, sgRNA2, sgRNA3, sgRNA4) (Fig. 1a and Supplementary Tables 1, 2). Among the sgRNAs used for the co-transfection of SNU719 and C666-1 cells, sgRNA3 induced the highest expression of *BZLF1* and Zta as well as another immediate early lytic protein, Rta, and the early lytic protein, BGLF4 (Fig. 1b, c). The high efficiency of sgRNA3 in inducing EBV lytic reactivation was further demonstrated in SNU719, C666-1, and C17 cells engineered using an inducible Casilio (iCasilio) activator system. The iCasilio system was delivered by transduction with lentiviral vectors constitutively expressing sgRNA3, HA-dCas9-2A-EGFP, and a piggyBac transposon containing the Tet-On 3xFLAG-PUFa-p65HSF transactivator (Supplementary Table 2). The treatment of stably transfected SNU719 and C17 cells with doxycycline (Dox) resulted in transactivator induction and the expression of Zta, Rta, and downstream lytic proteins (e.g., BGLF4, EA-D, VCA and gp350) in these EBV-positive epithelial cancer cells (Fig. 1d–f, Supplementary Fig. 2a). The supernatants of the treated cells were collected, and the presence of infectious EBV virions was demonstrated by the successful infection of Akata[EBV-negative] cells with EBV. Transcripts of EBV-encoded genes were detected in the re-infected Akata cells (Supplementary Fig. 2b). The findings confirmed the endogenous activation of *BZLF1* induced a complete lytic cycle and the production of infective EBV virions in SNU719 and C17 cells. As reported previously, we observed the induction of an abortive early lytic cycle pattern in C666-1 cells[6,14]. In stably transfected C666-1 cells treated with Dox, no late lytic proteins (VCA and gp350) were detected, whereas immediate early (Zta, Rta) and early (BGLF4) lytic proteins were induced (Fig. 1d–f). The endogenous activation of *BZLF1* showed significant in vitro cytolytic effects in EBVaGC and NPC cells. The Casilio-mediated artificial activation of endogenous *BZLF1* expression significantly inhibited the cell viability and colony formation ability of EBV-positive cancer cells (Fig. 1g, h). Strikingly, the potent cytotoxic effects of artificial activation of *BZLF1* were also demonstrated in vivo in nude mouse models implanted with iCasilio-engineered EBV-positive epithelial cancer cells. In mouse models of EBV-positive epithelial cancer, daily Dox treatment alone or in combination with GCV had a dramatic inhibitory effect on tumor formation (Supplementary Fig. 3a, b). These findings demonstrated that the artificial activation of endogenous *BZLF1* expression could efficiently reactivate the lytic cycle and induces a cytolytic effect in EBV-positive epithelial cancers.

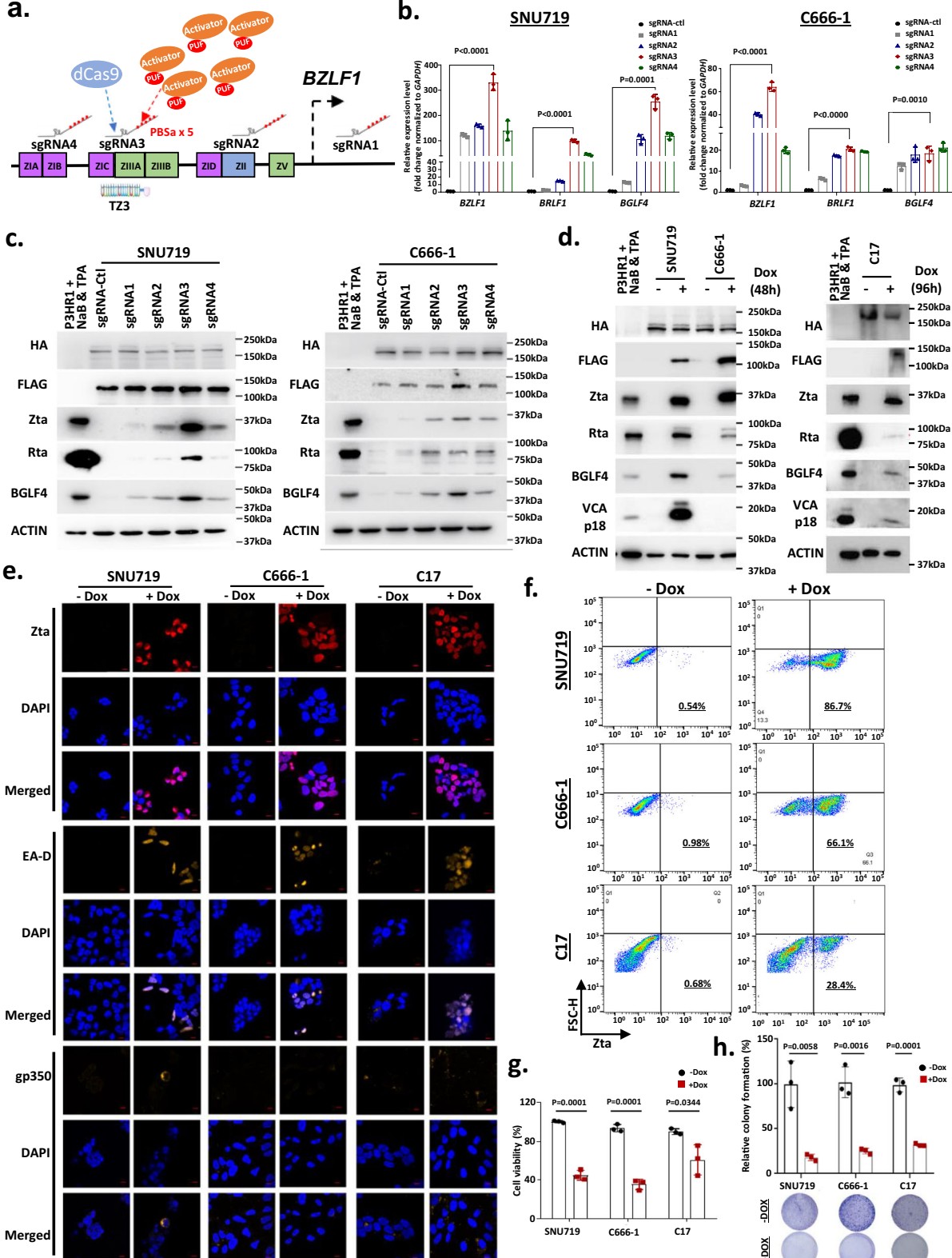

## Development of a TALE activator for the artificial activation of *BZLF1*

Using the well-established Casilio activation system with designed sgRNAs, we identified a target sequence in the *BZLF1* promoter for artificial activation, in support of the effort to develop a synthetic transcriptional activator for efficient EBV lytic induction therapy. However, it would be challenging to effectively deliver the complex Casilio activator system containing three complex and large constructs for the targeting of EBV-positive tumors in patients. Thus, we assembled a single construct encoding a precise and efficient TALE transcriptional activator that comprised only a central repeat domain to enable DNA recognition, nuclear localization signals (NLS), and a transcriptional activation domain to artificially activate *BZLF1* transcription (Fig. 2a). Based on the validated binding sequences of

**Fig. 1 | Reactivation of EBV lytic genes in EBV-associated cancers via CRISPR-based transcriptional activators. a** Schematics of the Casilio activator system targeting the *BZLF1* promoter in the EBV genome. The positions of the binding regions of the four sgRNA-5xPBSa (sgRNA1-4) in the *BZLF1* promoter are shown. Multimerization of the activator by the Casilio system enabled robust transcriptional activation of the endogenous *BZLF1* gene. Quantitative RT-PCR (**b**) and western blotting (**c**) demonstrated the endogenous expression of *Zta*/BZLF1, Rta/BRLF1 and BGLF4 in SNU719 and C666-1 cells at 24 h after transient transfection with HA-dCas9-2A-EGFP, 3xFLAG-PUFa-p65HSF1, and sgRNAs (*n* = 3 experimental replicates). Data are presented as mean ± SD. A *P* < 0.05 was considered to indicate statistical significance. **d** In SNU719, C666-1 and C17 cells stably transfected with sgRNA3, HA-dCas9-2A-EGFP, and inducible 3xFLAG-PUFa-p65HSF1 transactivator, the expression of Zta and other EBV lytic proteins (Rta, BGLF4 and VCAp18) were detected by western blotting after doxycycline (Dox) treatment for 48 h (*n* = 3 experimental replicates). **e** Immunofluorescence staining illustrated the expression of Zta, EA-D, and gp350 lytic proteins in Dox-treated stably transfected SNU719, C666-1, and C17 cells (scale bar = 10 μm) (*n* = 3 experimental replicates). **f** Using flow cytometry analysis, the percentage of Zta-expressing cells was determined in the stably transfected SNU719, C666-1, and C17 cells at 48 h post-Dox treatment. Representative flow cytometry plots are shown (*n* = 3 experimental replicates). **g** Inhibition of cell viability was detected in Dox-treated stably transfected SNU719, C666-1, and C17 cells using the CCK8 assay (*n* = 3 experimental replicates). Data are presented as mean ± SD. A *P* < 0.05 was considered to indicate statistical significance. **h** *BZLF1* induction significantly suppressed the colony formation ability of Dox-treated stably transfected EBV-positive cancer cells (*n* = 3 experimental replicates). Data are presented as mean ± SD. A *P* < 0.05 was considered to indicate statistical significance. **b, g, h** Two-sided student's t-test. Source data are provided as a Source Data file.

sgRNAs in the Casilio activator system, we designed and constructed a TALE activator plasmid, TZ3, that would specifically target the sgRNA3 binding sequence in the *BZLF1* promoter to activate endogenous Zta expression in EBV-positive tumor cells (Fig. 2a, Supplementary Tables 1–3). The constructed TALE transcriptional activator has an open reading frame of ~3.6 kb that encoded a FLAG tag and the designed TALE DNA-binding and NLS domains, fused to the p65HSF1 transactivator. In addition to TZ3, we also constructed TALE transcriptional activators that targeted regions overlapping with the binding sequences of sgRNA1, sgRNA2, and sgRNA4 (TZ1, TZ2, and TZ4, respectively) to determine the concordance of Casilio- and TALE-based artificial transcriptional activation systems. As shown in Fig. 2b, c, Zta expression was induced in SNU719 and C666-1 cells transiently transfected with *BZLF*-targeted TALE plasmids, but not in cells transfected with a control vector lacking the DNA-binding domain. In addition to Zta, the expression of downstream immediately early (Rta), early (BGLF4), and late (VCA) lytic proteins was detected in the EBV-positive SNU719 cells transfected with TZ3. Furthermore, TZ3 induced high levels of expression of immediate early (Zta, Rta) and early (BGLF4) lytic proteins in C666-1 cells. These findings confirmed that ectopic expression of the synthetic transcriptional activator TZ3 successfully reactivated the virus lytic cycle in EBV-positive tumor cells. Consistent with the findings obtained when using Casilio activator system with sgRNA3, TZ3 induced the highest level of Zta expression when compared with the other TALEs (TZ1, TZ2, and TZ4).

As shown in supplementary Fig. 4a, the TZ3-targeted sequence in the *BZLF1* promoter is conserved across all reported EBV variants[15]. Using electrophoretic mobility shift assay (EMSA) analysis, we showed the binding of TZ3 activator protein to the predicted *BZLF1* promoter sequence in the EBV genome, but not in the mutant probes (Supplementary Fig. 4b). These findings proved that the synthetic TZ3 protein specifically bound to the *BZLF1* promoter in the EBV genome to activate the transcription of this immediate early lytic gene. Furthermore, the transient transfection of HK1 cells, an EBV-negative NPC cell line, with TZ3 did not induce significant changes in the transcriptome or in the cell viability and cell cycle regulation processes (Fig. 2d–f). These findings demonstrated the high specificity of the synthetic TALE-transcriptional activator for targeting EBV-positive tumor cells. Next, TZ3 was used for the further development of TALE-based lytic induction therapy EBV-associated epithelial cancers.

## Synthesized nucleoside-modified mRNA encoding TZ3 lytic activator

To efficiently activate endogenous *BZLF1* expression for clinical application, we synthesized nucleoside-modified mRNAs encoding TALE-based transcriptional activators for in vivo delivery using LNPs. The T7 promoter in the TZ3 construct was used to initiate the in vitro transcription (IVT) of *TZ3* mRNA. For in vitro and in vivo studies, nucleoside-modified *TZ3* mRNAs with 5'-capping and a 3'-poly-A tail were synthesized and encapsulated by the formulated LNPs containing ALC-0315, 1,2-distearoyl-sn-glycero-3-phosphocholine (DSPC), cholesterol and 1,2-dimyristoyl-rac-glycero-3-methoxypolyethylene glycol-2000 (DMG-PEG200). These components have previously been used in LNPs formulated for the in vivo delivery of modified mRNAs in BNT162b2, a U.S. Food and Drug Administration (FDA)-approved SARS-CoV-2 vaccine[16]. Using dynamic light scattering analysis (DLS), the size of the formulated LNP encapsulated *TZ3* mRNA (mTZ3-LNP)s was determined to be ~124 nm, and its polydispersity index was 0.124 ± 0.032 (Fig. 3a). The encapsulation efficiency of the synthesized mTZ3-LNP was as high as 90% (Supplementary Fig. 5a, b). In Fig. 3b, we illustrate the uptake of mTZ3-LNPs by EBV-positive SNU719 cells and the intracellular release of the *TZ3* mRNAs using confocal laser scanning microscopy.

After 24 h of incubation with mTZ3-LNP, efficient induction of Zta expression was detected in EBV-positive SNU719 and C666-1 cells by western blotting (Fig. 3c). The induction of Zta expression was observed in SNU719 and C666-1 cells at 12 and 18 h post-mTZ3-LNP treatment respectively (Fig. 3d). Using quantitative reverse transcription-polymerase chain reaction (qRT-PCR), we revealed the upregulation of *BZLF1* transcripts in SNU719 and C666-1 cells as early as 4 and 8 h post-mTZ3-LNP treatment, respectively (Supplementary Fig. 6). The induction of EBV immediately early (Rta), early (BGLF4, EA-D) and late (VCA) lytic proteins downstream of Zta was observed in these cells at later time points (Fig. 3d). The early lytic proteins BGLF4 and EA-D were expressed in C666-1 cells at 48–72 h post-mTZ3-LNP treatment. In SN719 cells, the expression of early (BGLF4, EA-D) and late (VCA) lytic proteins was induced at 18 h post-mTZ3-LNP treatment (Fig. 3d). The prevalence of SNU719 and C666-1 cells expressing Zta and downstream early lytic protein EA-D was demonstrated by immunofluorescence staining. Expression of the late lytic protein gp350 was found in the SNU719 cells at 96 h post-mTZ3-LNP treatment (Fig. 3e). Flow cytometry analysis detected up to 71.9% and 83.7 % of Zta-positive tumor cells in mTZ3-LNP treated SNU719 and C666-1 cells respectively (Fig. 3f). Compared with Zta-positive cells, smaller proportions of EA-D- and gp350-positive cells were detected in the mTZ3-LNP treated SNU719 and C666-1 cells at 72 and 96 h post-treatment indicated the heterogenous status of lytic cycle in these lytic reactivated EBV-positive cancer cells. Notably, at 72–96 h post-treatment, caspase-3 cleavage was induced in mTZ3-LNP-treated EBV-positive cancer cells, including C666-1 cells that experienced only abortive lytic cycle reactivation. These findings demonstrated that the synthesized mTZ3-LNPs produced functional TZ3 transcriptional activators that could induce *BZLF1* expression in EBV-positive cancer cells.

In addition to SNU719 and C666-1, the ability of mTZ3-LNP to reactivate Zta expression was tested in a panel of EBV-positive cancer models, including four NPC (C17, NPC43, NPC43-M81, and NPC76c), two EBVaGC (AGS-EBV and YCCLE1) and two BL (Akata-EBV and P3HR1) cell lines (Fig. 4). Strikingly, mTZ3-LNP efficiently induced Zta expression in all EBV-positive cancer cell lines. After 48 h of treatment with mTZ3-LNP, flow cytometry analysis revealed proportions of

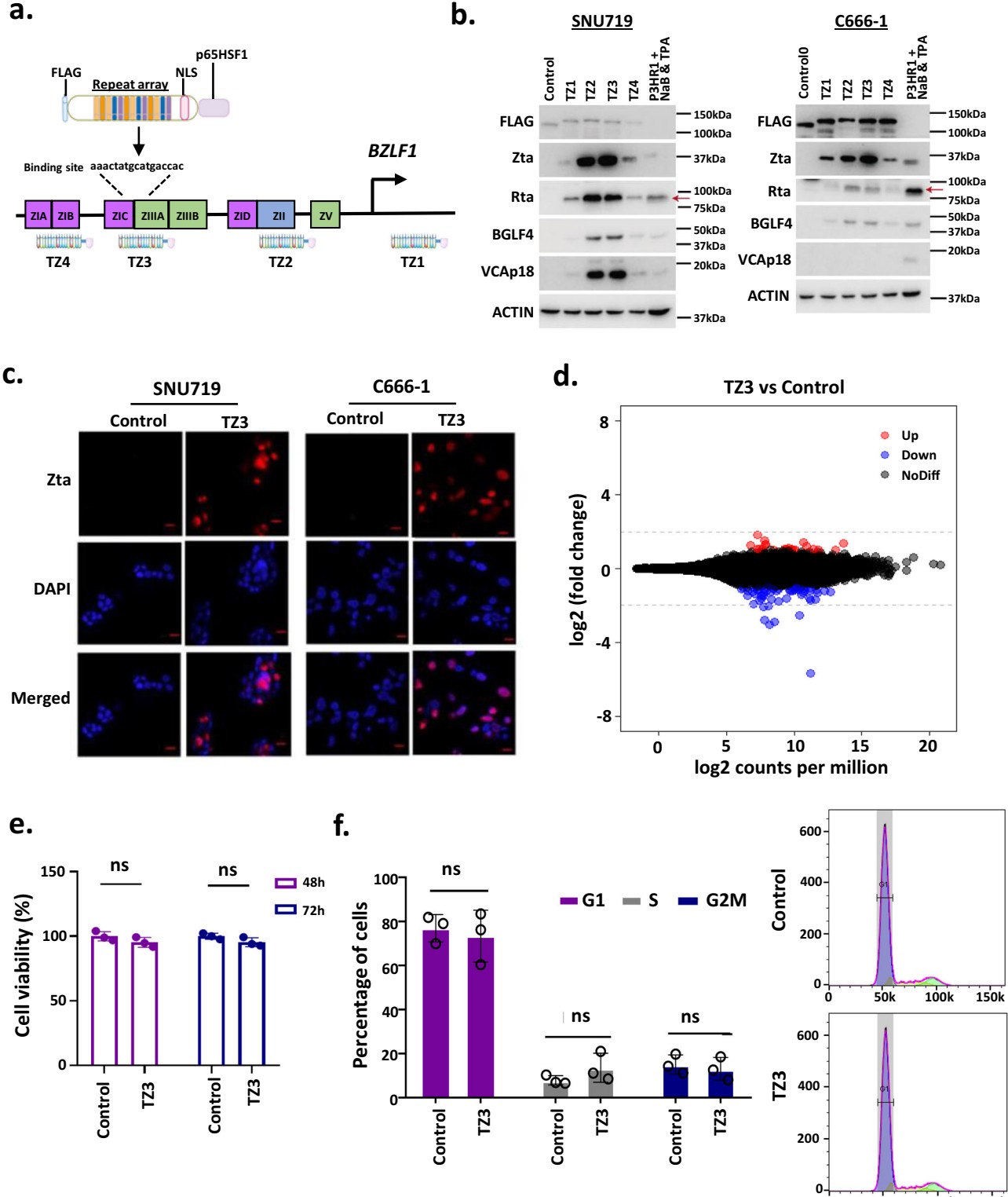

**Fig. 2 | Synthetic *BZLF1*-specific TALE transcriptional activator. a** Schematics of the TALE transcriptional activator targeting the *BZLF1* promoter in the EBV genome. **b** Western blotting detected the endogenous expression of immediate early (Zta, Rta), early (BGLF4), and late (VCAp18) lytic proteins in SNU719 and C666-1 cells transiently transfected with the designed *BZLF1*-targeted TALE plasmid, TZ3 and other TALE plasmids, TZ1, TZ2 and TZ4. P3HR1 cells treated with NaB and TPA were included as the controls for lytic protein expression (*n* = 3 experimental replicates). **c** Using immunofluorescence (IF) staining, Zta expression was detected in SNU719 and C666-1 cells transiently transfected with TZ3 (*n* = 3 experimental replicates).

Representative IF images are shown. Scale bar = 10 μm. **d** Using RNA sequencing, differentially expressed genes between TZ3-transfected and control HK1 cells were determined (*n* = 3 experimental replicates). Few genes showed significant changes in expression in TZ3-transfected HK1 cells. **e** No significant reduction of cell viability was detected in the TZ3-transfected HK1 cells (*n* = 3 experimental replicates). Data are presented as mean ± SD. ns not significant. **f** Ectopic transfection of TZ3 did not significantly alter the cell cycle of HK1 cells (*n* = 3 experimental replicates). Data are presented as mean ± SD. ns not significant; **e, f** Two-sided student's t-test. Source data are provided as a Source Data file.

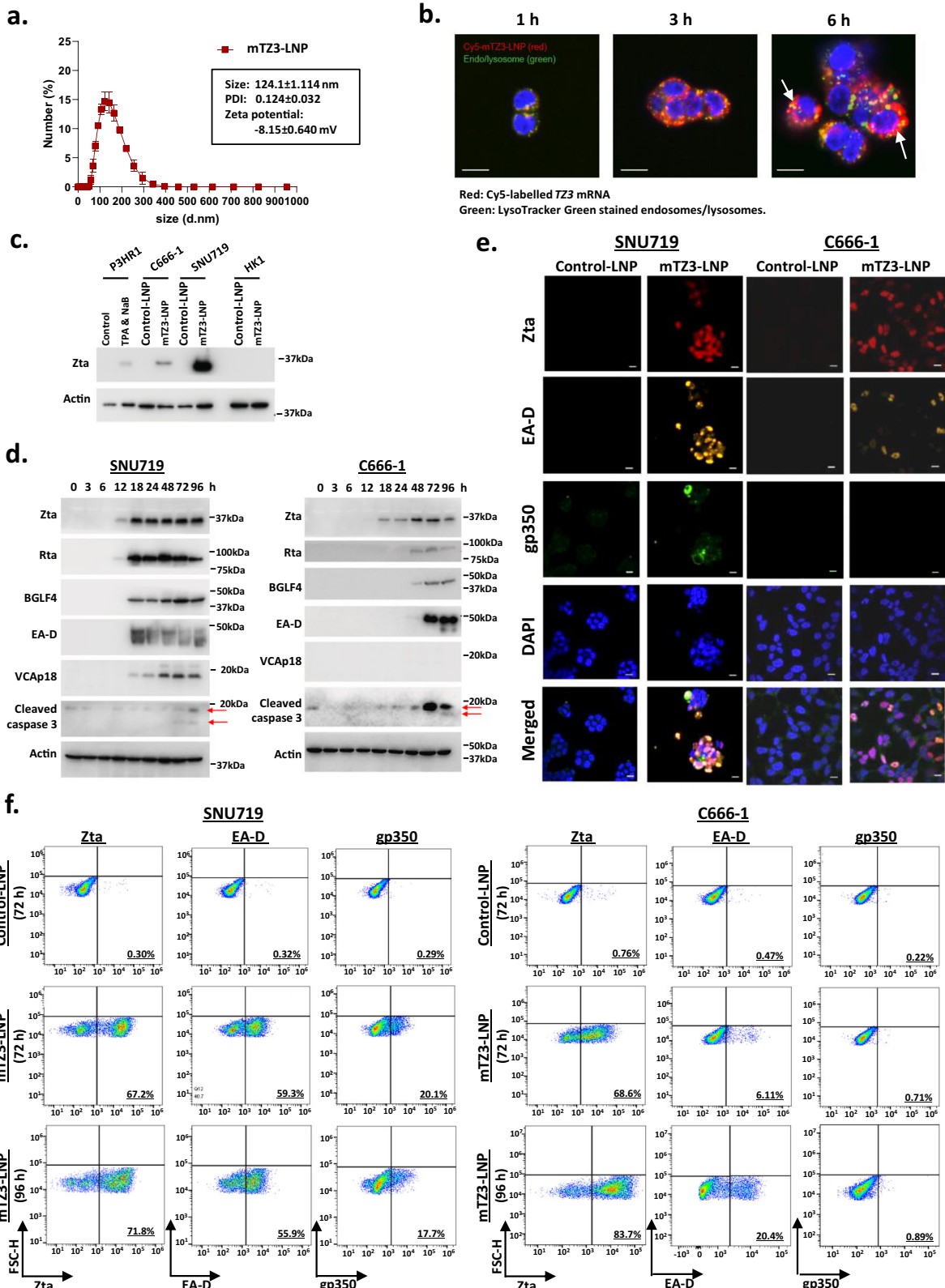

Zta-expressing cells ranging from ~26.4–92.2% in this panel of EBV-positive cancer models (Fig. 4a–d). The induced expression of Zta and its downstream EBV lytic proteins was also demonstrated in mTZ3-LNP-treated YCCLE1 and NPC43-M81 cells using western blotting (Fig. 4e). Compared with chemical activators, mTZ3-LNP reactivated the EBV lytic cycle in EBV-positive epithelial cancers much more efficiently. At 48 h post-mTZ-LNP treatment, we detected Zta expression

in up to 86.2% and 78.9% of SNU719 and C666-1 cells, respectively, compared with only 19.5% and 3%, respectively, after sodium butyrate (NaB) treatment (Fig. 4a, b). Our study thus demonstrated that mTZ3-LNP is a potent lytic activator for different types of EBV-positive malignant cells.

Using RNA-sequencing (RNA-seq), we revealed that few cellular genes were transcriptionally activated in EBV-negative HK1 cells

**Fig. 3 | Reactivation of EBV lytic genes by LNP-encapsulated nucleoside-modified mRNA (mTZ3-LNP). a** The particle size, polydispersity index, and zeta potential of mTZ3-LNP were analyzed using a dynamic light scattering method. The data are representative of four independent experiments. Data are presented as mean ± SD. **b** The cell uptake process of the LNP-encapsulated Cy5-labeled mTZ3 mRNAs by SNU719 cells over 1, 3 and 6 h was visualized using an LSM 880 confocal laser scanning microscope (*n* = 3 experimental replicates). The fluorescence signals were measured in three channels: Cy5, excitation/emission wavelength (ex/em), 633/697 nm; Dnd-26, ex/em, 488/524 nm; and hoechst, ex/em 405/460 nm. Scale bar = 10 μm. The signal of Cy5-labeled mTZ3 mRNAs in the cytoplasm of SNU719 cells is indicated by a white arrow. **c** Western blotting was used to detect Zta

expression in SNU719 and C666-1 cells treated with mTZ3-LNPs and LNP-encapsulated control mRNA (control-LNP) for 24 h. The EBV-negative NPC cell line HK1 was used as the negative control (*n* = 3 experimental replicates). **d** Western blotting detected the expression of Zta, Rta, the downstream early (BGLF4) and late (VCA) lytic proteins and cleaved caspase 3 in SNU719 and C666-1 cells treated with mTZ3-LNPs for 3 to 96 h (*n* = 3 experimental replicates). **e** Immunofluorescence staining and (**f**) flow cytometry analysis illustrated the induction of Zta, EA-D, and gp350 expression in mTZ3-LNP-treated SNU719 and C666-1 cells (*n* = 3 experimental replicates). Representative IF images and flow cytometry plots are shown. Scale bar = 10 μm. Source data are provided as a Source Data file.

treated with mTZ3-LNP, indicating the high specificity of the TZ3 transcriptional activator (Fig. 5a, b). We observed a similar result in HK1 cells transiently transfected with the TZ3 construct (Fig. 2d). Notably, the significant upregulation of EBV and other cellular transcripts was observed in mTZ3-LNP-treated SNU719 and C666-1 cells (Fig. 5b). In addition to *BZLF1*, EBV transcriptome profiling demonstrated the abundant expression of multiple lytic gene transcripts (e.g., *BRLF1, BGLF4, BXLF1, LF3, BALF2, BHLF1, BMRF1*) in EBV-positive tumor cells treated with mTZ3-LNP (Supplementary Fig. 7 and Supplementary Data 1, 2). Furthermore, mTZ3-LNP mediated *BZLF1* expression and lytic reactivation induced the expression of multiple cellular genes in EBV-positive SNU719 and C666-1 cells (Supplementary Data 1, 2). Despite the differentially expressed genes involved in multiple cellular mechanisms, few of them were detected in both mTZ3-LNP-treated SNU719 and C666-1 cells (Supplementary Fig. 8). These distinct transcription patterns may be attributable to unique genomic changes in each cell line and the abortive lytic cycle in C666-1 cells.

In addition to the RNA-seq study, we confirmed the absence of an off-target effect of mTZ3-LNP treatment via chromatin immunoprecipitation (ChIP)-sequencing analysis with an anti-FLAG antibody. As shown in supplementary Fig. 9, the predicted TZ-binding sequences in the *BZLF* promoter from the EBV genome were enriched in mTZ3-LNP-treated EBV-positive SNU719 cells. ChIP-sequencing analysis did not identify any potential TZ3-binding sequences in the exon, intron, and regulatory regions of human genes. Furthermore, the specific cytotoxicity of the synthetic mTZ3-LNP for EBV-positive cancer cells is also demonstrated in Fig. 5c. Specifically, a significant reduction in cell viability was observed in EBV-positive SNU719 and C666-1 cells treated with mTZ3-LNP. Notably, mTZ3-LNP neither alone nor in combination with GCV exerted a significant effect on the viability of the EBV-negative NPC cell line, HK1.

**Reactivation of EBV lytic genes by in vivo delivery of mTZ3-LNP**

The in vivo delivery of mRNA to EBV-positive tumors via the formulated ALC-0315-LNPs was examined in nonobese diabetic severe combined immunodeficiency disease (NOD-SCID) mouse models. For this purpose, luciferase mRNA-encapsulated LNPs were intravenously injected into mice. Notably, luciferase protein signals were detected in the tumors at 24 h after injection (Supplementary Fig. 10a). No luciferase protein signals were found in other normal organs of the mice except for the liver. Through intravenous injection of Dil C18-labeled LNP encapsulated *TZ3* mRNA into NOD-SCID mice, we determined that the half-life of the formulated mTZ3-LNPs in circulation was 8.34 h (Supplementary Fig. 10b). Dil C18 fluorescent signals were also detected in the tumor tissue from the mice at 3 h post-injection of Dil C18-labeled LNP encapsulated TZ3 mRNA (Supplementary Fig. 10c).

The efficient induction of endogenous Zta expression in EBVaGC and NPC tumors in vivo was demonstrated by the intravenous injection of mTZ3-LNP into NOD-SCID mice implanted with EBV-positive cancer cells. Using immunohistochemical staining, we revealed the obvious induction of Zta, EA-D/BMRF1 and gp350 expression in tumor cells from SNU719 xenografts at 12, 24, and 48 h post-injection of mTZ3-LNP; however, we did not observe similar findings in the controls

(Fig. 6a). Expression of Zta, EA-D and gp350 lytic proteins was detected in 10%, 12%, and 9% of tumor cells, respectively, at 48 h post-treatment. The findings indicated that, in vivo, mTZ3-LNP treatment successfully induced the expression of the early and late lytic proteins in EBV-positive tumor xenografts in NOD-SCID mouse models. In addition to *BZLF1*, the abundant transcription of other downstream lytic genes including *BGLF4, BMRF1, and BLLF1* was detected in mTZ3-LNP treated tumors via RNAscope RNA in situ hybridization assays (Fig. 6c). Various amounts of immediate early or early lytic gene transcripts were observed in the tumor cells, indicating heterogeneity in terms of the lytic cycle stage. Our study demonstrated the transcriptional activation of *BZLF1, BGLF4, BMRF1,* and *BLLF1* genes in 15.1%, 12.7%, 18.3%, and 8.1% of tumor cells, respectively, at 48 h post-intravenous injection of mTZ3-LNP (Fig. 6d). Notably, *BGLF4* is not only a significant marker of lytic cycle progression but also encodes a serine/threonine kinase that converts the non-cytotoxic GCV to a cytotoxic form.

**In vivo therapeutic efficacy of mRNA-based lytic induction therapy**

The therapeutic efficacy of combined mTZ3-LNP and GCV treatment for EBV-associated epithelial cancer was evaluated in NOD-SCID mouse models implanted with SNU719, C666-1, and C17 cells and an NPC patient-derived xenograft (PDX), Xeno-76. mRNA-based lytic induction treatment was started when the tumors sizes reach 80–100 mm³. mTZ3-LNP was injected through the tail vein every 2–3 days, whereas GCV was injected intraperitonially every day (Fig. 7a). As shown in Fig. 7b–d and Supplementary Figs. 11a, 12, 13, potent growth inhibition of SNU719, C666-1, C17, and xeno-76 tumors was observed in mice treated with mTZ3-LNP alone or in combination with GCV compared with controls (*P* < 0.005). Similar tumor growth inhibitory effects of mTZ3-LNP alone and in combination with GCV treatment were observed in all EBV-positive xenograft models. No obvious changes in the body weights of the mice were observed during treatment (Supplementary Fig. 11b). Notably, the residual tumors harvested from mice with mTZ3-LNP treatment alone or in combination with GCV contained markedly fewer tumor cell components but increased proportions of necrotic lesions, lymphocytes and fibroblasts (Fig. 7b, Supplementary Fig. 12). The increased inflammation may be induced by EBV lytic reactivation and ongoing cell death process although NOD-SCID mice lack of B and T lymphocytes. This observation reinforced our observation of the potent in vivo antitumor effects of mTZ3-LNP against EBV-positive cancers. In addition to the absence of changes in body weight, mice treated with mTZ3-LNP and GCV exhibited neither tissue damage to the organs nor significant changes in the serum concentrations of alanine aminotransferase (ALT), aspartate aminotransferase (AST) and creatinine, highlighting the safety of mTZ3-LNP-mediated lytic induction therapy in preclinical models of EBV-associated cancer (Supplementary Fig. 11b and 14a–c).

## Discussion

The persistent latency in EBV-positive epithelial cancer cells is tightly controlled by acquired genetic changes and epigenetic modifications

 

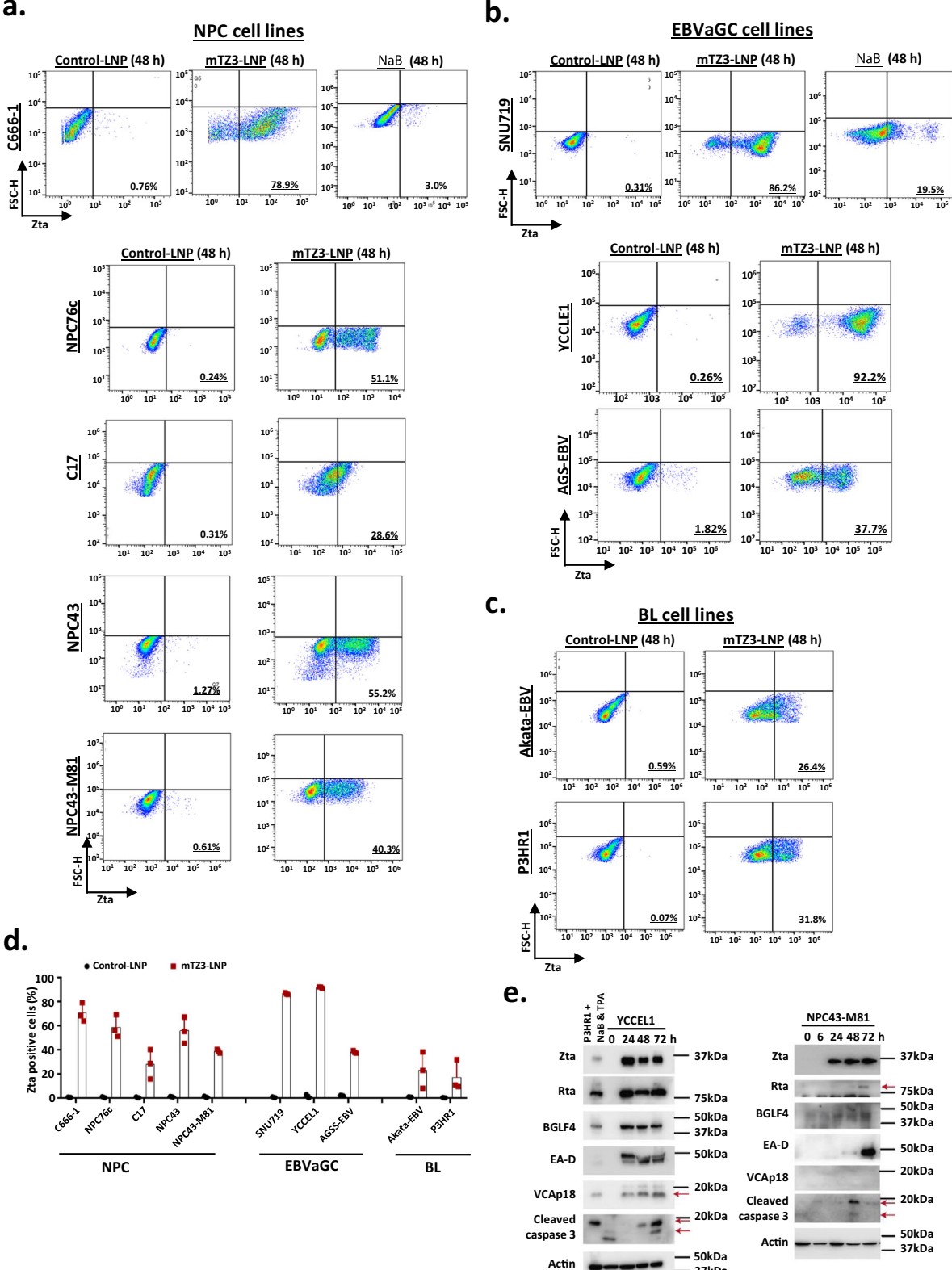

**Fig. 4 | Highly efficient EBV lytic reactivation in a panel of EBV-positive cancer cells treated with mTZ3-LNPs.** Representative flow cytometry plots show the high efficiency of mTZ3-LNP treatment (48 h) for inducing Zta expression in a panel of EBV-positive tumor cell lines, including (**a**) NPC (C666-1, NPC76c, C17, NPC43 and NPC-M81), (**b**) EBVaGC (SNU719, YCCLE1 and AGS-EBV) and (**c**) Burkitt lymphoma (P3HR1 and Akata-EBV). SNU719 and C666-1 cells treated with NaB were included as a reference of chemically induced lytic reactivation (*n* = 3 experimental replicates).

**d** The percentages of Zta-positive cells in the NPC, EBVaGC and BL cell lines treated with mTZ3-LNP for 48 h are shown (*n* = 3 experimental replicates). Data are presented as mean ± SD. **e** The expression of Zta, its downstream lytic proteins (Rta, BGLF4, and EA-D), and cleaved caspase 3 in YCCEL1, NPC31M81, and C17 cells treated with mTZ3-LNPs were detected by western blotting (*n* = 3 experimental replicates). Source data are provided as a Source Data file.

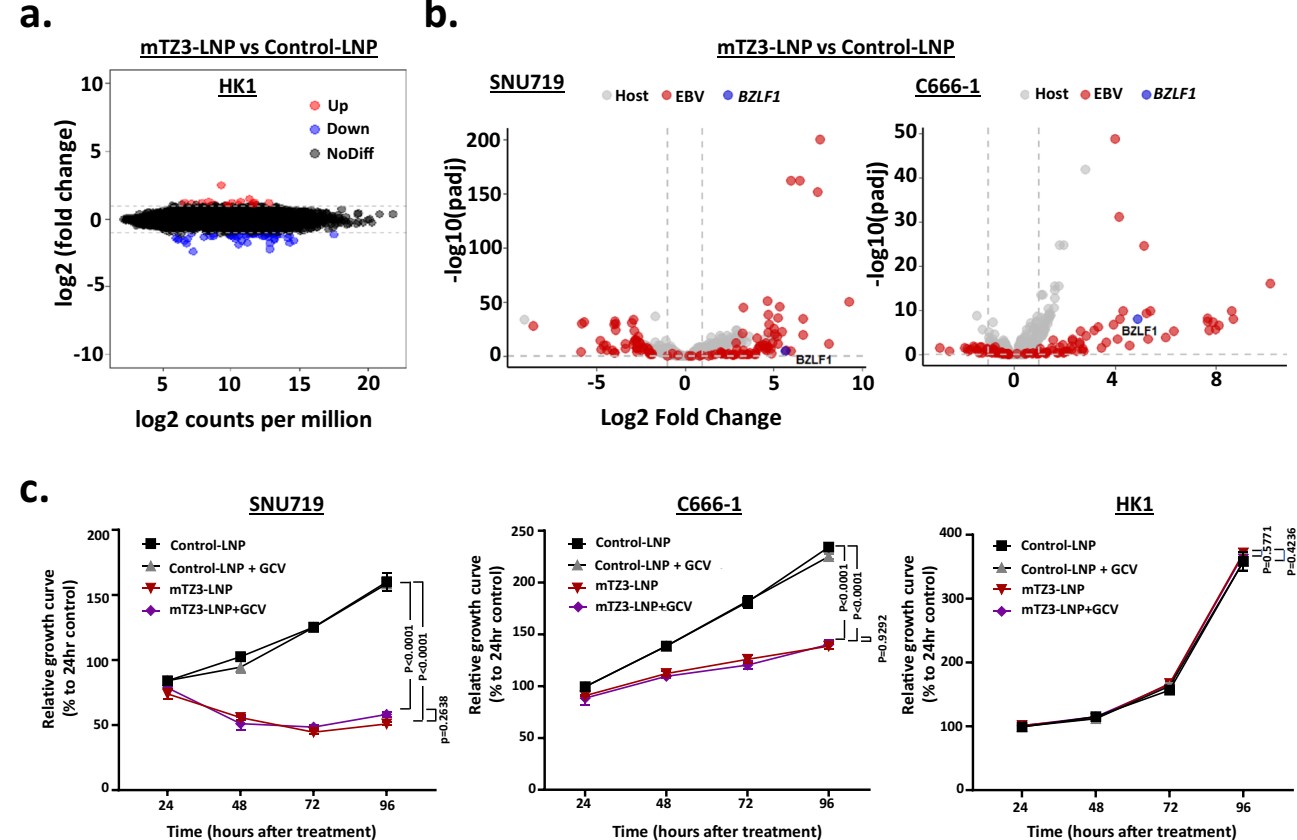

**Fig. 5 | Effects of mTZ3-LNP treatment on EBV-positive epithelial cancer cells.**
**a** Using RNA sequencing, a few significantly differentially expressed genes were detected in EBV-negative HK1 cells treated with mTZ3-LNP ($n = 3$ experimental replicates). **b** Differentially expressed genes identified in SNU719 and C666-1 cells treated with mTZ3-LNP versus those treated with control mRNA-LNP for 48 h ($n = 3$ experimental replicates). *BZLF1* and EBV-encoded transcripts are illustrated in blue and red dots, respectively. **c** The viability of SNU719, C666-1, and HK1 cells treated with mTZ3-LNP alone and a combination of mTZ3-LNP and GCV for 24, 48, 72, and 96 h was determined ($n = 3$ experimental replicates). Significant growth inhibition was observed in the EBV-positive SNU719 and C666-1 cells treated with mTZ3-LNP alone and in combination with GCV. Data are presented as mean ± SD. A $P < 0.05$ was considered to indicate statistical significance. One-way analysis of variance (ANOVA). Source data are provided as a Source Data file.

in the viral and host genomes. Multiple viral and cellular factors have been shown to regulate the latent-lytic switch to prevent cell death by lytic induction during transformation and clonal expansion[7–9]. Over the past few decades, reports have described various drugs or reagents with lytic induction capacity in EBV-positive tumor cells. These chemical lytic inducers include histone deacetylase inhibitors (e.g., sodium butyrate, valproic acid, suberanilohydroxamic acid, and romidepsin), DNA methyltransferase inhibitors (e.g., 5-aza-2'-deoxycytidine and 5-azacytidine), protein kinase C activators (e.g., TPA), chemotherapeutics agents (e.g., gemcitabine), antibacterial antibiotic (e.g., clofoctol) and several novel compounds (e.g., C7 and E11). These compounds reactivate EBV lytic genes through different mechanisms, targeting either epigenetic regulation or cellular signaling pathways[8]. Although phase-I/II clinical trials of CLVA therapy with a combination of gemcitabine, valproic acid, and valganciclovir treatment have been shown it to be safe and to elicit clinical responses in patients with recurrent NPC, response rates of less than 30% have been reported[10,11]. Previous in vitro studies have shown that the chemical lytic inducers elicit cell-context- and cell-type-specific lytic induction responses from EBV-positive epithelia cancer cells. Most chemical lytic inducers reactivate EBV lytic genes much less efficiently in patient-derived EBV-positive epithelial cancer cell lines (e.g., SNU719, C666-1, NPC43, and C17) than in EBV re-infected cancer cells (e.g., HK1-EBV, AGS-BX1, and HONE-1-EBV) (Supplementary Fig. 1)[17]. The lytic induction response in EBV-infected tumor cells is believed to be influenced by epigenetic modifications and aberrant oncogenic signaling pathways acquired

during clonal expansion. In addition to its low efficiency at lytic reactivation and intertumoral heterogeneity, the clinical implementation of chemical lytic inducers for EBV lytic induction therapy has been limited by the broad-spectrum cytotoxicity of these compounds. EBV lytic reactivation cannot be induced effectively at low doses of these drugs. At high doses, however, both EBV-infected and un-infected normal cells may be indiscriminately killed by the drugs.

To overcome the high level of complexity of EBV latent-lytic switch regulatory mechanisms, we developed a TALE-based transcriptional activator to artificially activate the expression of the EBV immediate early gene *BZLF1* in EBV-positive tumor cells (Fig. 8). This LNP-encapsulated mRNA encoding a *BZLF1*-specific TALE-transcriptional activator directly targets the *BZLF1* promoter in EBV genome. Neither the mRNA nor transcriptional activator is chemical probe that interacts with the molecules regulating the latent-lytic switch in EBV. By exploiting a highly specific LNP-encapsulated mRNA encoding a TALE transcriptional activator, we demonstrated the potent in vivo antitumor effects of the artificial activation of endogenous *BZLF1* expression in multiple models of EBV-positive epithelial cancer. We further demonstrated the highly specific cytotoxicity of this approach against to EBV-positive cells. The synthetic transcriptional activator was shown to specifically target the *BZLF1* promoter in the EBV-positive cancer cells and did not induce transcriptional activity in EBV-negative cells. In addition to its reported growth arrest and cytotoxic effects, Zta can upregulate the transcription of various cellular genes contributing to multiple cancer hallmarks. Our approach

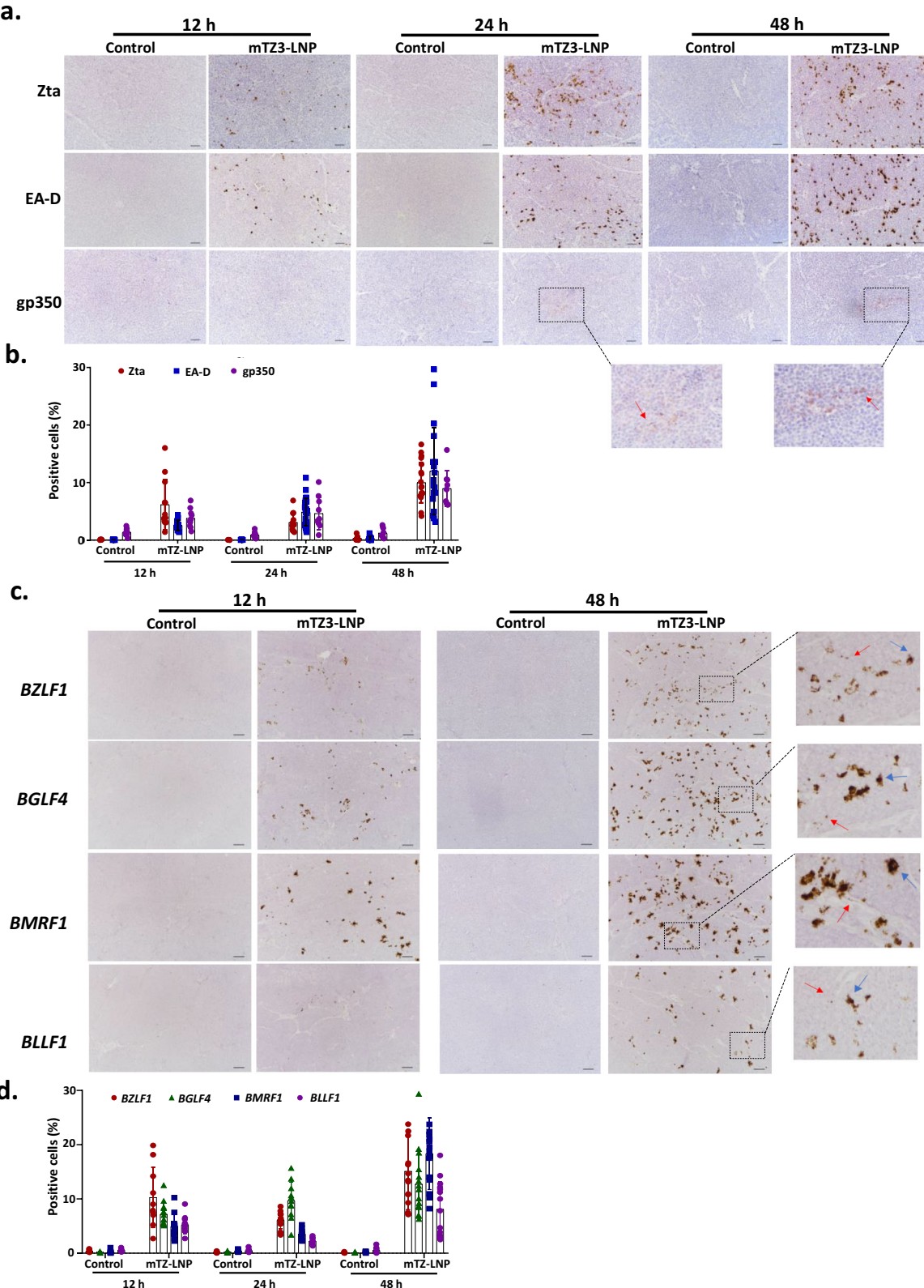

avoids the safety issues caused by the cytotoxicity and potentially oncogenic effects of ectopic Zta expression in non-infected cells[18].

Unlike the Casilio CRISPR-Casilio activator platform, which has three essential components, the in vivo delivery of a TALE transcription activator to primary and metastatic tumors of the patients using LNP-encapsulated mRNA technology is feasible. We exploited the LNP formulation used in an FDA-approved mRNA vaccine to develop a mRNA drug, mTZ3-LNP, for highly efficient lytic induction therapy against EBV-associated epithelial cancers. In our study, the safety and specificity of mTZ3-LNP were demonstrated in in vitro and in vivo preclinical models of EBVaGC and NPC. In addition to its capability for specific binding to the targeted sequence, the strong transcriptional activity of the TALE transcriptional activator, TZ3, may be due to the high copy number of EBV episomes and the hypomethylated *BZLF1*

**Fig. 6 | mTZ3-LNP reactivates EBV lytic gene expression in in vivo EBV-positive tumor models. a** Using immunohistochemical staining, the expression of Zta, EA-D, and gp350 was detected in the representative tissue sections of SNU719 tumors from NOD-SCID mouse models at 12, 24, and 48 h after the intravenous administration of mTZ3-LNP ($n = 3$ mice/group). The tumor cells with gp350 expression were illustrated by red arrows. Scale bar = 50 μm. **b** The percentages of tumor cells with Zta, EA-D, and gp350 expression in the tumors from the mice were determined at 12, 24, and 48 h post-intravenous administration of mTZ3-LNPs. At least four different representative fields (×200 magnification) obtained from the section of each triplicate were counted ($n$ = at least 4 fields/tumor examined over 3 mice). Data are presented as mean ± SD. **c** The expression of EBV immediate early (*BZLF1*),

early (*BMRF1* and *BGLF4*), and late (*BLLF1*) lytic gene transcripts in representative FFPE sections of SNU719 tumors from NOD-SCID mouse models at 12 and 48 h post-treatment with mTZ3-LNP was determined using RNAscope RNA in-situ hybridization ($n = 3$ mice/group). Representative tumor cells expressing high and low copy numbers of EBV lytic gene transcripts are indicated by blue and red arrows respectively. Scale bar = 50 μm. **d** The percentages of tumor cells expressing *BZLF1*, *BGLF4*, *BMRF1*, and *BLLF1* mRNAs in the tumors from the mice were determined at 12, 24, and 48 h post-treatment with mTZ3-LNP. At least four different representative fields (×200 magnification) obtained from the section of each triplicate were counted ($n$ = at least 4 fields/tumors examined over 3 mice). Data are presented as mean ± SD. Source data are provided as a Source Data file.

promoter in EBV-positive tumor cells[7–9]. Importantly, once its expression is induced by the mTZ3-LNP, Zta can either cis- or trans-activate the *BZLF1* promoters of multiple EBV episomes in tumor cells. Zta also drives the expression of *BRLF1*, which encodes the transactivator Rta; in turn, Rta induces *BZLF1* transcription, forming a positive feedback loop to activate the *BZLF1* promoter. Both mechanisms maintain the activation of *BZLF1* and the accumulation of Zta after the transient expression of synthetic TZ3 transcriptional activators in EBV-positive tumor cells[9]. In C17, an NPC cell line containing only 2–3 copies of the EBV genomes per cell, treatment with artificial activators also induced expression of Zta. As shown in the Supplementary Fig. 2, the delayed transcriptional activation of *BZLF1* and downstream lytic genes was observed in C17 cells after Dox treatment. This finding suggests that tumor cells with a low copy number of EBV genome copy number require a longer period of Zta accumulation to activate the expression of downstream early and late lytic proteins. Notably, a significant growth inhibitory effect was observed in C17 CDX models in vivo following the administration of multiple mRNA-TZ3 doses. In addition, the high efficiency of mTZ3-LNP in inducing lytic reactivation in different types of EBV-associated cancers is attributable to its mechanism of action, which is not dependent on the host's epigenetic status or aberrant signaling pathways.

Using a panel of tumor xenograft models, our in vivo study highlighted the therapeutic efficacy of mRNA-based lytic induction therapy against EBV-positive epithelial cancers. The safety of the intravenous administration of mTZ3-LNP for long-term treatment was demonstrated in NOD-SCID mouse xenograft models. For clinical implementation of lytic induction therapy, co-administration with GCV is essential for the rapid and specific killing of EBV-positive tumor cells and inhibition of the infective virion production. Through in vivo study, we observed no significant difference in growth inhibition between tumors treated with mTZ3-LNP alone or in combination with GCV. The absence of an obvious bystander effect with combined treatment may be due to the high efficiency of lytic reactivation and the potent cytotoxic effect of mTZ3-LNP on EBV-positive tumors. Despite of the limited bystander killing effect observed, the GCV administration is an essential procedure for rapid and specific killing of the EBV-positive cells and inhibit the production of infective virions during EBV lytic induction treatment. In addition to the direct cytotoxic effects, the potent innate and adaptive immune responses induced by abundant immunogenic lytic proteins may contribute to the effective eradication of EBV-positive cancers during lytic induction therapy[19,20].

A limitation of this study is that we were unable to determine the responses of both the tumor microenvironment and the host immune system to tumor cells induced to undergo EBV lytic reactivation in our immunodeficient mouse models. The NOD-SCID mouse model lacks B- and T-lymphocytes, which are key immune cells triggering the host's immune responses to the highly expressed EBV lytic antigens. Nevertheless, we observed the NK cells accumulated in the adjacent necrosis regions and infiltrated into the residue tumors in the mTZ3-LNP treated mouse models (Supplementary Fig. 14). In a future study, we will

establish EBV-positive tumor xenografts in a humanized mouse model, allowing us to accurately elucidate the innate and adaptive immune responses induced by mTZ3-LNP treatment and the potential therapeutic effects of combined treatment with mTZ3-LNP and immune checkpoint blockade[21]. Combination of mTZ3-LNP with immunotherapeutic strategies such as immune checkpoint blockade and NK cell therapy may further enhance the treatment response of patients with EBV-associated cancers.

Similar to highly successful COVID-19 vaccines used in clinic, the development of mRNA therapies against a wide range of human diseases, including cancers, is on the rise. In this proof-of-concept study, we successfully developed a first-in-class mRNA drug for lytic induction therapy against EBV-associated epithelial cancers. By exploiting nucleoside-modified mRNA technologies, non-viral delivery strategies, and the TALE artificial activator system, we produced mTZ3-LNP, a highly efficient inducer of the lytic reactivation and selective killing of EBV-positive tumor cells. This mRNA nanomedicine provides a promising clinical opportunity for lytic induction therapy against EBV-associated epithelial cancers.

## Methods

### Cell lines and patient-derived xenografts

The EBVaGC cell line SNU719 was obtained from the Korean Cell Line Bank, Seoul, Republic of Korea. The EBVaGC cell lines YCCEL1 and AGS-EBV were provided by Professors Qian Tao and Jun Yu from the Chinese University of Hong Kong respectively[22,23]. The EBV-positive NPC cell lines C666-1, C17, NPC43, NPC43-M81 and NPC76c were established in our laboratory[24–26]. The cell lines were used in various in vitro experiments. Xeno-76, an EBV-positive NPC PDX established from a NPC patient in Hong Kong, was used in our in vivo study[26]. The collection and use of the NPC specimens for the establishment of the Xeno-76 PDX were approved by the Institutional Review Board of the University of Hong Kong and the patient's consent was obtained as described in previous reported study[26]. All animal care and experimental procedures for using the NPC PDX were approved by the University Animal Experimentation Ethics Committee (AEEC), the Chinese University of Hong Kong. The EBV-negative NPC cell line HK1 was included as a control[27]. Two EBV-positive BL cell lines, P3HR1 and Akata-EBV, maintained in our laboratories were also used in the study. Except for C17, NPC43, NPC43M81, and NPC76c, all of the cells were maintained in Roswell Park Memorial Institute (RPMI)-1640 medium (Sigma, St. Louis, MO, USA), supplemented with 10% fetal bovine serum (Gibco, Waltham, MA, USA). To maintain the growth of the C17 and NPC76c cell lines, 0.5 μM of Y-27632 (Enzo Life Sciences Inc., Farmingdale, NY, USA), a ROCK inhibitor, was added to the RPMI-1640 medium. All cell cultures and all biological experiments were performed at 37 °C under 5% $CO_2$. All of the cell lines used in this study were authenticated using short tandem repeat (STR) profiling and *EBER* in situ hybridization when we started the related experiments and prepared this report. All cells were tested for mycoplasma contamination by PCR using the primer set 5′-YGCCTGVGTAGTAYR-YWCGC-3′ (MYCO5) and 5′-GCGGTGTGTACAARMCCCGA-3′ (MYCO3).

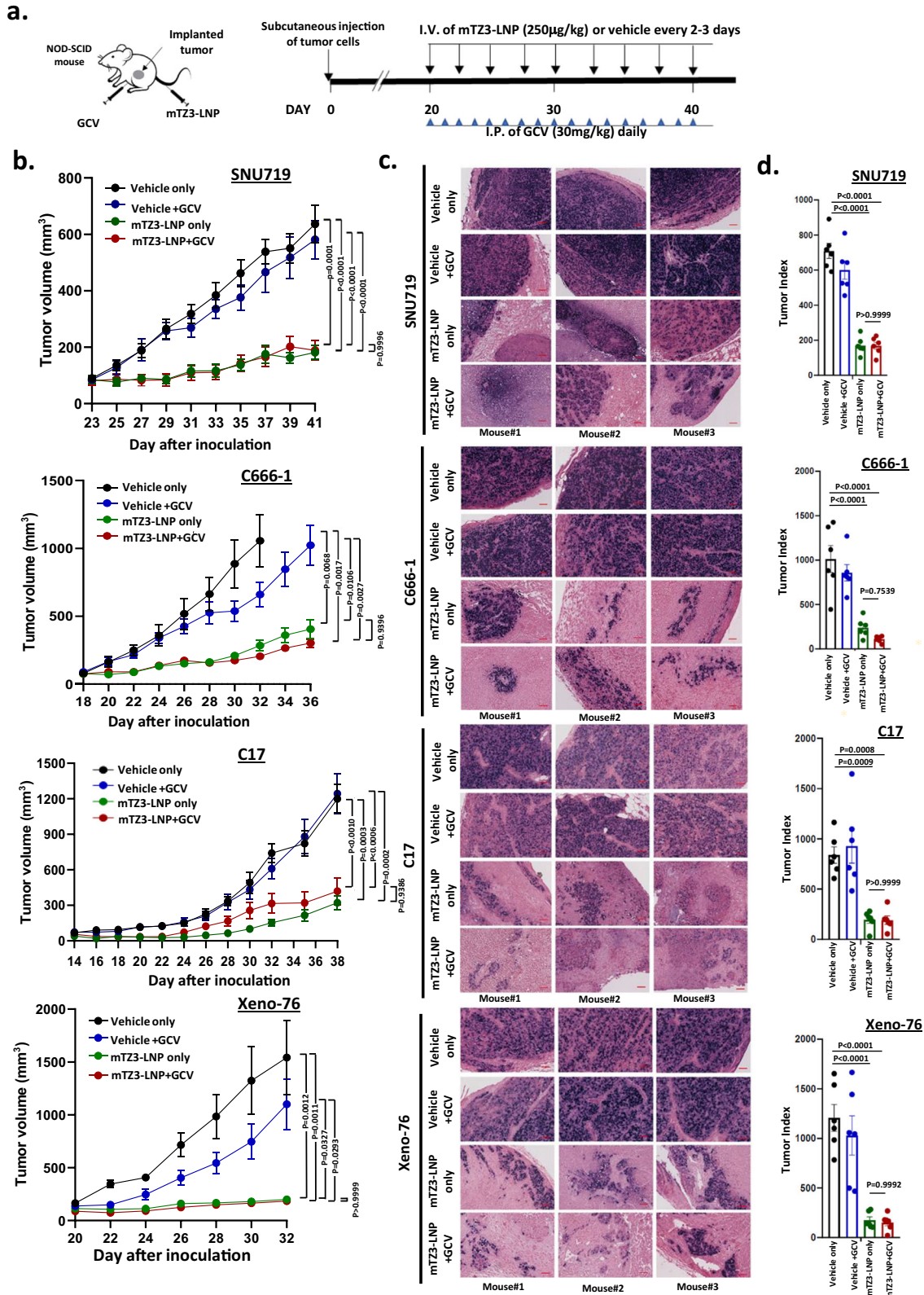

## Casilio activator system targeting *BZLF1*

Casilio, a hybrid system combining CRISPR/dCas9 and the Pumilio RNA-binding domain fused effector was used to induce *BZLF1* transcription with the activator module 3xFLAG-PUFa-p65HSF1 and a specific sgRNA with five copies of the Pumilio binding site. The sgRNA is appended with five copies of Pumilio binding sites with the sequence UGUAUGUA that can each recruit one molecule of cognate PUFa RNA binding domain fused to a hybrid activator composed of p65 and HSF1 activation domains to the dCas9 target site, thus achieving robust

**Fig. 7 | In vivo inhibition of EBV-positive epithelial cancers by mTZ3-LNP-based lytic induction treatment. a** A scheme illustrating the in vivo treatment of EBV-positive EBVaGC (SNU719) and NPC (C666-1, C17, Xeno-76) preclinical xenograft NOD-SCID mouse models with mTZ3-LNP and GCV. **b** Tumor volumes were measured throughout the treatment period (SNU719 and C666-1: $n = 7$ mice/group; C17 and Xeno-76: $n = 6$ mice/group). Data are presented as mean ± SEM. A $P < 0.05$ was considered to indicate statistical significance. **c** Using *EBER* in-situ hybridization, EBV-positive tumor cells were detected in representative FFPE sections of residual tumors harvested after treatment. Scale bar = 250 μm. Representative images from

$n = 6$–7 mice/group are shown. **d** The tumor index of each harvested EBV-positive tumor was determined after treatment with mTZ3-LNP alone, combined mTZ3-LNP and GCV, GCV, and vehicle controls. Tumor index = tumor volume × percentage of the *EBER*-positive area. Significant tumor growth inhibition was observed in SNU719, C666-1, C17, and Xeno-76 xenografts treated with mTZ3-LNP alone or combined with GCV (SNU719 and C666-1: $n = 7$ mice/group; C17 and Xeno-76: $n = 6$ mice/group). Data are presented as mean ± SD. ns not significant; A $P < 0.05$ was considered to indicate statistical significance. One-way analysis of variance (ANOVA). Source data are provided as a Source Data file.

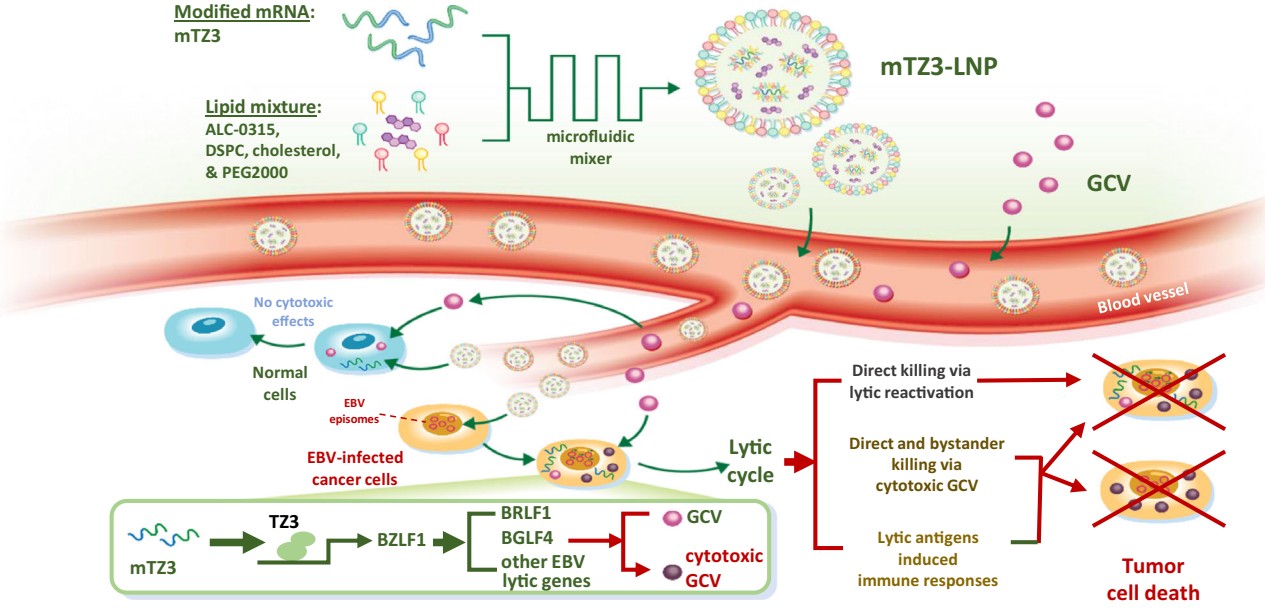

**Fig. 8 | mRNA-LNP-based lytic induction therapy targets EBV-positive epithelial cancers.** Schematic illustration of the synthesis of mTZ3-LNP and its application in lytic induction therapy against EBV-positive epithelial cancers. The LNP-encapsulated modified mRNA mTZ3-LNP encodes the *BZLF1*-specific TALE transcriptional activator TZ3 to induce the transcription of EBV lytic genes in EBV-positive tumor cells. Treatment induces the lytic cycle to kill EBV-infected cells directly and activate GCV cytotoxicity and bystander-killing effects in the tumor. mTZ3-LNP treatment may also activate the host's innate and adaptive immune responses to EBV-positive cancers.

activation of the target gene[12]. The lentiviral vectors used in this study to constitutively express HA-dCas9-2A-EGFP, 3xFLAG-PUFa-p65HSF1, and four BZLF1-targeting sgRNA-5xPBSa (sgRNA1-4) used in this study are listed in Supplementary Table 1. For the inducible Casilio activator system, the 3xFLAG-PUFa-p65HSF1 transactivator was subcloned downstream of a Tet-On promoter on a piggyBac transposon vector that constitutively expressed rtTA (Supplementary Table 1).

### Construction of TALE plasmids
*BZLF1* promoter-targeted TALEs were designed and constructed in vitro using the Golden Gate assembly protocol (Golden Gate TAL Effector Kit 2.0, #1000000024; Addgene, Watertown, MA, USA) as described previously[28]. Based on the identified target sequences on the *BZLF1* promoter, the amino acid sequences of the binding domain of TZ3 and other TALEs (TZ1, TZ2, and TZ4) were designed as shown in Supplementary Tables 2, 3. The constructed TALEs were then fused with the p65 activation domain to enable transcription. In brief, a DNA sequence designed to express FLAG-NLS-lacZ-p65HSF1 was first cloned into the pcDNA3.1 vector. Customizable polymorphic amino acid repeats from the TAL effectors targeting the EBV BZLF1 promoter sequence were used to replace the lacZ using multiple rounds of Golden Gate cloning. The TZ3 plasmid or the other TALE plasmids (TZ1, TZ2, and TZ4) was either transiently transfected into the cells for expression or linearized for IVT.

### Preparation of nucleoside-modified mRNA
The TZ3 plasmid was linearized by *XmaI* digestion and purified. The linearized TZ3 plasmid was then used as the template under the regulation of the T7 promoter. The nucleoside-modified mRNA was synthesized as described[29]. In brief, in vitro transcription was conducted using a T7 high-yield RNA synthesis kit (New England BioLabs, Ipswich, MA, USA), with 100% replacement of UTP with N1meΨTP and 1 μg of template. The reactions were incubated at 37 °C for 4 h, followed by treatment with RNase free DNase I. Next, mRNA capping was performed using the Vaccinia Capping System (New England BioLabs), followed by the addition of 3'poly(A)-tails using *E. coli* poly(A) polymerase (New England BioLabs). mRNA purification was performed using the Monarch RNA Cleanup Kit (New England BioLabs).

### Preparation and characterization of LNP-encapsulated mRNA
mTZ3-LNPs were prepared by mixing an ethanol phase containing lipids with purified mTZ3 mRNA in an aqueous phase in a microfluidic device. In brief, the ethanol phase was prepared by solubilizing a mixture of the ionizable lipid ALC-0315 (Cayman Chemical, Ann Arbor, MI, USA), 1,2-distearoyl-sn-glycero-3-phosphocholine (DSPC; Avanti, Alabaster, AL, USA), cholesterol (Sigma) and 1,2-dimyristoyl-rac-glycero-3-methoxypolyethylene glycol-2000 (DMG-PEG 2000; Avanti) at a molecular ratio of 50:10:38.5:1.5 in ethanol. The aqueous phase was prepared by diluting mTZ3 mRNA or luciferase mRNA (TriLink, San

Diego, CA, USA) eight-fold in 200 mM acetate buffer (pH 5.0). Syringe pumps were used to mix the aqueous and ethanol phases at a ratio of 3:1. The resulting LNPs were dialyzed against 1× phosphate-buffered saline (PBS) in a 20,000 MWCO cassette (Invitrogen, Carlsbad, CA, USA) at 4 °C for 2 h. The encapsulation efficiency of the mRNA-LNPs was calculated as reported previously[29]. In brief, the samples were treated with PBS buffer alone (as unencapsulated mRNA) or with 2% Triton X-100 (as total mRNA). The Qubit RNA HS Assay Kit (Invitrogen) was used, and the RNA concentrations were measured using a Qubit 4 Fluorometer (Invitrogen). The encapsulation efficiency (EE) was calculated using the following formula: EE = [1 − (unencapsulated mRNA/ total mRNA) × 100%][30,31]. The average size, polydispersity index, and Zeta potential of the formulated mTZ3-LNP were determined using a dynamic light scattering method and the Zetasizer Nano ZS90 system (Malvern Panalytical, Worcestershire, UK). The samples were diluted with PBS before measurement. To investigate cell uptake of the formulated mTZ3-LNP, the LNP-encapsulated Cy5-labeled mTZ3 mRNAs were incubated with the SNU719 cells for 1, 3, and 6 h. The fluorescent signals emitted by Cy5-mTZ3-LNP and LysoTracker Green (Invitrogen) stained endosomes were measured using an LSM 880 confocal laser scanning microscope with the AxioObserver system (Zeiss, Jena, Germany). The fluorescence signals were measured in three channels: Cy5, excitation/emission wavelength (ex/em), 633/697 nm; Dnd-26, ex/em, 488/524 nm; and Hoechst, ex/em 405/460 nm.

### Quantitative real-time PCR
Total RNA was extracted from the cells using TRIzol reagent (Thermo Fisher Scientific, Waltham, MA, USA). The extracted RNA was then reverse transcribed into complementary DNA using the RT Regent Kit with gDNA Eraser (TaKaRa, Kyoto, Japan) and quantitative real-time PCR was performed using SYBR Green master mix (Thermo Fisher Scientific). The mRNA expression levels of the EBV lytic genes were normalized to those of *GAPDH*. The primer sequences are listed in Supplementary Table 4.

### Western blot assay
Protein extracts were prepared using RIPA lysis buffer supplemented with protease inhibitor (Roche, Basel, Switzerland). The protein concentrations were determined using a protein assay reagent (Bio-Rad, Hercules, CA, USA) against a bovine serum albumin (BSA) standard curve. Equal amounts of proteins in each extract were separated by sodium dodecyl-sulfate polyacrylamide gel electrophoresis (SDS-PAGE) and transferred onto 0.45 μm nitrocellulose membranes. The blocked membranes were incubated with the appropriate primary antibodies. The primary antibodies used in this study include anti-BZLF1/Zta (BZ1, Santa Cruz Biotechnology, Dallas, TX, USA; 1:1000), anti-Rta (8C12, Argene, Varilhes, France;1:1000), anti-EA-D (1108-1, Santa Cruz Biotechnology; 1:1000); anti-BGLF4 (1:1000), VCAp18 (#PA1-73003, Invitrogen; 1:500); anti-cleaved caspase 3 (Asp175, Cell Signaling; 1:1000) and anti-Actin (13E5, Cell Signaling; 1:4000) antibodies. After being washed and incubated with secondary antibody. Signals in the blots were detected using the ChemiDoc Image system (Bio-Rad).

### Fluorescence-activated cell sorting (FACS) analysis
The cells treated with chemical inducers, mTZ3-LNP or control-LNP were trypsinized, collected, and washed with cold PBS. The cells were then fixed in freshly prepared 4% paraformaldehyde and permeabilized with 0.1% TritonX-100 in PBS. The cell pellets were then stained with an Alexa-647-conjugated mouse anti-BZLF1/Zta antibody (BZ1, Santa Cruz Biotechnology; 1:100), Alexa-594-conjugated anti-EA-D antibody (1108-1, Santa Cruz Biotechnology; 1:100) or Alexa-488-conjugated anti-gp350 antibody (Santa Cruz Biotechnology; 1:100) and analyzed using BD LSRFortessa Cell Analyzer (Becton Dickinson, Franklin Lakes, NJ, USA). The data

were analyzed using FlowJo software, version 10 (FlowJo, LLC, Ashland, OR, USA).

### Immunofluorescence staining
For immunofluorescence staining, the cells were seeded onto coverslips in 6-well plates the day before treatment. The cells were then washed with PBS, fixed in 4% paraformaldehyde, and permeabilized with 0.1% Triton X-100 in PBS for 30 min. The cells were then incubated with the Alexa-647 conjugated anti-BZLF1 antibody (BZ1, Santa Cruz Biotechnology; 1:100), Alexa-594 conjugated anti-EA-D (1108-1, Santa Cruz Biotechnology; 1:100) or Alexa-488 conjugated anti-gp350 (0221, Santa Cruz Biotechnology; 1:100) for 2 h at room temperature in the dark. Finally, the stained cells were counterstained with DAPI and mounted on slides with Dako fluorescence mounting medium (Agilent, Santa Clara, CA, USA). The images were processed using an LSM 880 confocal laser-scanning microscope with Zen software (Zeiss, Oberkochen, Germany).

### Immunohistochemical staining
The expression of the Zta, EA-D, and gp350 proteins in the tumor sections of EBV-positive xenografts was detected using immunohistochemical staining. In brief, 4 μm sections were obtained from tumors grown in mice treated with PBS, GCV, mTZ3-LNP or a combination of mTZ3-LNP and GCV. The paraffin-embedded sections were dewaxed, rehydrated, and washed with water. After antigen retrieval, the samples were incubated with an anti-BZLF1/Zta (BZ1, Santa Cruz Biotechnology; 1:100), anti-EA-D (1108-1, Santa Cruz Biotechnology; 1:100) or anti-gp350 (0221, Santa Cruz Biotechnology; 1:100) primary antibodies. The sections were then incubated with a horseradish peroxidase-labeled secondary antibody, developed with 3,3′-diaminobenzidine, and counterstained with hematoxylin (Sigma). The percentages of cells expressing EBV lytic proteins in the tumor sections from mice treated with mTZ3-LNP and controls were evaluated. Representative images were acquired using a Nikon ECLIPSE Ni-E microscope equipped with a Ds-Ri2 microscope camera and NIS-Elements software (Nikon, Tokyo, Japan). At least four different pictures obtained in each triplicate (×200 magnification) and analyzed using ImageJ software to determine the percentage of tumor cells expressing EBV lytic proteins.

### *EBER* in situ hybridization
EBV-positive cancer cells were detected in tumor specimens using by *EBER* in situ hybridization assay. An *EBER* probe ISH kit (Leica, Newcastle, U.K.) was used to confirm the presence of EBV in formalin-fixed paraffin-embedded (FFPE) tumor sections, according to the manufacturer's instructions.

### RNAscope RNA in situ hybridization
The expression of *BZLF1* and a panel of lytic gene transcripts (*BMRF1*, *BGLF4*, and *BLLF1*) were detected using RNAscope 2.0 RISH assays and a panel of EBV lytic gene-specific probes (Advanced Cell Diagnostics, USA) The formalin-fixed paraffin-embedded (FFPE) tissue sections (5 μm) were deparaffinized in xylene and then dehydrated in a series of ethanol in water. The *BZLF1*-, *BMRF1*-, *BGLF1*- or *BLLF1*-specific probes were hybridized to the sections, followed by incubation with pre-amplifier, amplifier, and labeled probes according to the manufacturer's instruction[26]. Representative images were acquired from the stained sections. The percentages of cells expressing EBV lytic transcripts (*BZLF1*, *BMRF1*, *BGLF1*, and *BLLF1*) in the tumor sections from mice treated with mTZ3-LNP and controls were evaluated as described in the immunohistochemical staining sections.

### Detection of infectious EBV particles
Culture supernatants from EBV-positive tumor cells induced to undergo artificial EBV lytic reactivation were harvested and

centrifuged at 260 g for 5 min and then filtered through a 0.45 μm cellulose acetate filter to remove the cell debris. The centrifuged and filtered supernatants, which contained the EBV particles, were further subjected to ultracentrifugation at 27,400 g for 4 h at 4 °C to pellet the EBV particles. The supernatant was discarded, and the pellet was resuspended in RPMI-1640 medium supplemented with 10% fetal bovine serum at a volume 1/30 of the original supernatant volume. These procedures increased the EBV concentration in the culture supernatant by 30×. EBV-negative Akata cells were then incubated with the concentrated EBV supernatant for 3 days. The infected Akata cells were then harvested and subjected to DNA and RNA extraction to detect the presence of the EBV genome and EBV gene expression, respectively[25].

## RNA-sequencing

To evaluate the RNA profiles of the cancer cells after TZ3 transfection or mTZ3-LNP treatment, total RNA was extracted from the cells using TRIzol reagent (Invitrogen). RNA sequencing libraries were prepared using the Swift RNA Library Kit (Swift Biosciences, Ann Arbor, MI, USA) with DNase I treatment and rRNA and globin depletion. Next-generation sequencing (150 bp, paired-end) was performed using an Illumina HiSeq1500 sequencing system (Illumina, San Diego, CA, USA). The adapter sequences and low-quality sequences in the total sequencing reads were filtered before downstream analysis. In brief, the reads were mapped, aligned, and annotated to the human reference genome (GRCh38) and EBV genome (chrEBV_Akata_inverted) using Hisat2 (2.1.0) with the "--rna-strandness RF" parameter and StringTie (1.3.6)[32]. Downstream analyses were performed on the R software platform (v4.1.0). Differentially expressed genes between the control and TZ3-treated samples were identified using DEseq2 (1.32.0) with the criterion of a false discovery rate below 0.05[33]. The expression levels of protein-coding genes were further determined via gene set enrichment analysis (GSEA), using Hallmark and GO: BP gene sets obtained from the Molecular Signature Database and the clusterProfiler package (4.0.5)[34,35]. Volcano plots were generated using the ggplot2 package (3.4.1). The raw data in the fastq files of the RNA-sequencing have been deposited in Sequence Read Archive (SRA) on National Center for Biotechnology Information (NCBI) under accession number PRJNA1007461.

## Chromatin immunoprecipitation sequencing

The genomic specificity of the TALE transcriptional activator TZ3 was evaluated in mTZ3-LNP-treated C666-1 cells using ChIP-sequencing, as described previously[36]. mTZ3-LNP-treated and control cells were fixed in 1% formaldehyde and quenched by glycine. Chromatin was prepared using a truCHIP Chromatin Shearing Kit (Covaris, Woburn, USA) and broken into 100–500 bp fragments using Covaris S220 Focused-ultrasonicator (Covaris). The protein–DNA complexes were immuno-precipitated using 5 μg of an anti-FLAG antibody (F1804, Sigma) on a rotator at 4 °C overnight and then purified using magnetic beads (26162, Pierce; Thermo Fisher). After washing, crosslink reversal and DNA purification were performed, and 8 ng of immunoprecipitate and input DNA were used for each Illumina sequencing library construction according to the manufacturer's protocol (Kapa Hyper Prep Kit, KK8504, Roche). Each library was sequenced on a Nextseq 500 platform (Illumina) to obtain 150 base paired-end reads. The sequencing tags were mapped against the Akata reference genome (accession no. KC207813) using Bowtie 2. Unique FLAG-tag mapper tags were used for broad peak calling by MACS2 analysis. The raw data in the fastq files of the ChIP-sequencing have been deposited in SRA on NCBI under accession number PRJNA1007461.

## EMSA

EMSA was performed to determine the binding of TZ3 transactivator to its target sequence in the *BZLF1* promoter in the NPC cell line C666-1[37]. In addition to the wild-type sequence, three mutant sequences were included to demonstrate the binding specificity. The oligonucleotides (WT probe, Mutant Probe-1, Mutant Probe-2, and Mutant Probe-3) were labeled with biotin using the Biotin 3' End DNA Labeling Kit (Thermo Fisher Scientific) (Supplementary Fig. 4b). Five pmol of the oligonucleotide were added to 25 μl of ultrapure $H_2O$, 10 μl of 5X Terminal Deoxynucleotidyl Transferase (TdT) Reaction Buffer, 5 μl of 5 μM Biotin-11-UTP, and 5 μl of TdT (2 U/μl), to a reaction volume of 50 μl. The reaction mixture was incubated at 37 °C for 30 min, then 2.5 μl of 0.2 M EDTA was added to stop the reaction. Fifty μl of choloroform:isoamyl alcohol was added to each reaction mixture to extract the TdT. The mixture was vortexed and centrifuged for 2 min at 14,000 G, then the top aqueous phase was withdrawn and saved. Binding reaction was performed by mixing 2 μl of 10× Binding Buffer, 1 μl of 1 μg/μl Poly(dI-dC), 1 μl of 1 μM biotin labeled oligonucleotides, 4 μg of nuclear protein extracts and $H_2O$, to a total reaction volume of 20 μl. For competitive experiments, 200 times molar excess of unlabeled oligonucleotides was added in addition of the other reaction components. Reaction mix was incubated at room temperature for 20 min. Separation was performed on 5% native PAGE gel in a SE600 Cooled Vertical Electrophoresis Unit (Hoefer). Transfer to positively charged nylon membranes (GE Healthcare) were performed in a Trans-Blot Cell (BioRad). Subsequently the proteins were crosslinked by UV irradiation and the membrane was developed using the Chemiluminescent Nucleic Acid Detection Module (Thermo Scientific).

## Cell viability determination

Approximately $10^4$ cells per well per 100 μL were seeded in 96-well plates the day before transfection with the TZ3 plasmid or treatment with mTZ3-LNP and GCV. For mTZ3-LNP treatment, 100 ng of mRNA per well was added to the 96-well plate the next day, with or without 10 μg/mL GCV in a total volume of 100 μL. At the end of the treatment, the medium was refreshed and 10 μl of CCK-8 reagent (Dojindo Molecular Technologies, Rockville, MD, USA) was added to each well to determine the cell viability. After incubation at 37 °C for 3–4 h, the absorbance was measured at 450 nm and 650 nm using a 96-well SpectraMax plate reader (Molecular Devices, San Jose, CA, USA). The cell growth inhibition in each well was calculated as follows: $(viability_{control} - viability_{drug})/viability_{control} \times 100\%$. Each sample was analyzed in triplicate.

## Cell cycle analysis

The cells were detached from the culture plates using trypsin, washed with cold PBS, and fixed in 70% ethanol at 4 °C overnight. The cells were then washed with PBS and incubated with propidium iodide (1 μg/mL; Invitrogen, P3566) and RNase (10 μg/mL; Roche) for 30-min. After washing, $10^4$ cells per sample were analyzed using a FACSCalibur Flow Cytometer (BD Biosciences) to detect the DNA content. FlowJo software was used for data analysis.

## In vivo mouse experiments

Minced Xeno-76 PDX tumor tissues or $5 \times 10^6$ SNU719, C666-1 or C17 cells were subcutaneously inoculated into the flanks of 5–6-week-old NOD-SCID mice with an initial body weight of -18–22 g; the tumors were allowed to grow to -100 mm³. The mice were housed in the following conditions: temperature of 20–23 °C, relative humidity of 40–60%, and a 12-h light/dark cycle (7:00 a.m.–7:00 p.m.). The mice were randomly assigned to different experimental groups and intravenously injected with either the vehicle (PBS) or mTZ3-LNP every 2 days for 2 weeks. GCV was intraperitoneally injected into the mice in the GCV only group and the combined mTZ3-LNP and GCV group. The mice were weighed, and their tumors were measured with a caliper every 3 days. When the tumor sizes exceeded 1000 mm³, the mice were

killed, and tumor and blood samples were collected for analysis. The maximal tumor size permitted by the University Animal Experimentation Ethics Committee (AEEC), the Chinese University of Hong Kong is 2000 mm$^3$. In the experiments, the maximum permitted tumor burden (2000 mm$^3$) was not exceeded in the mice at any time point. The tumor volume was calculated using the formula $0.5 \times l \times w^2$, where $l$ and $w$ represent the tumor length and width, respectively. Serum samples and organs, including the heart, lung, liver, spleen, and kidney, were collected at the end of the experiment to evaluate the in vivo cytotoxicity of the mTZ3-LNP treatment. The serum ALT, AST, and creatinine concentrations were measured. Formalin-fixed paraffin-embedded sections of the organs were subjected to hematoxylin and eosin (H&E) staining. The histological features were assessed by a pathologist (KF To).

To evaluate the circulation lifetime of the LNP-encapsulated mTZ3 mRNA in circulation in NOD-SCID mouse models, LNPs were fluorescently labeled with Dil C18 (Invitrogen). Four mice per group were intravenously injected with either Dil C18-labeled LNP-encapsulated mTZ3 mRNA at a dose of 0.5 mg RNA/kg or PBS as a control. Fifty microliters of blood were collected in the EDTA-treated tubes from the facial vein at 0, 8, 24, and 48 h post-injection. The circulating LNPs were measured by detecting the Dil C18 fluorescent signals in the blood samples, and the plasma was extracted. The circulating LNP were measured by detecting the Dil C18 signals in the plasma using a SpectraMax plate reader (Molecular Devices)[38]. All animal care and experimental procedures were approved by the University Animal Experimentation Ethics Committee (AEEC), the Chinese University of Hong Kong.

### Statistics & reproducibility

All graphs were generated using GraphPad 8 software (GraphPad Inc, San Diego, CA, USA), and all statistical analyses were performed using one-way analysis of variance (ANOVA) or two-sided Student's t-tests. All in vitro experiments were performed in triplicate. No data were excluded from the analyses. Error bars indicate the standard deviation (S.D.) unless otherwise specified to represent the standard error of the mean (SEM). A $P < 0.05$ was considered to indicate statistical significance.

### Reporting summary

Further information on research design is available in the Nature Portfolio Reporting Summary linked to this article.

## Data availability

The raw data in the fastq files of the RNA-sequencing and ChIP-sequencing have been deposited in Sequence Read Archive (RA) on National Center for Biotechnology Information (NCBI) under accession number PRJNA1007461. All remaining data can be found in the Article, Supplementary, and Source data files. Source data are provided with this paper.

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

## Acknowledgements

K.W.L. was supported by the Innovation and Technology Fund (Midstream Research Programme for Universities - MRP/036/21X); Research Grant Council, Hong Kong (Areas of Excellence Scheme—AoE/M-401/20; Collaborative Research Fund—C4001-18GF; General Research Fund—14101721), Health and Medical Research Fund (08191046), and Core Utilities of Cancer Genome and Pathobiology of the Chinese University of Hong Kong. L.L. was supported by the research fund to the Center for Neuromusculoskeletal Restorative Medicine from Health@InnoHK program launched by Innovation and Technology Commission, Hong Kong SAR. C.M.T. was supported by the General Research Fund (14113620, 14114523, 17122420), Early Career Scheme (24114922), Health and Medical Research Fund (09203176), and Faculty Innovation Award (FIA2020/A/01) of the Chinese University of Hong Kong.

## Author contributions

K.W.L., A.W.C. M.W., P.M.H. and L.L. led the study, designed and conducted the experiments, analyzed the data, interpreted the results, wrote the initial draft of the manuscript, and created the figures. C.M.T., Y.Y., A.T., G.T.Y.C., E.L.K., M.S.A.C., S.Y.H., W.C.T., N.J., J.J.Z. and H.H.Y.L. were involved in designed and conducting the experiments, developed the methodologies, helped draft the manuscript and approved the final version of the manuscript. C.M.T., J.Y.K., B.B.Y.M., M.R.C., C.L. and K.F.T. supervised the research, provided material or funding support, helped draft the manuscript, and approved the final version of the manuscript.

## Competing interests

K.W.L., A.W.C., P.M.H. and M.W. have filed a patent application "Activation of lytic genes in cancer cells" (WO 2021/173977A9; PCT/US2021/019880); K.W.L., M.W., P.M.H. and C.M.T. have filed a US provisional patent application, "Synthetic mRNAs for treatment of EBV-associated diseases" (no. 63/462,197). K.W.L., C.M.T. and M.W. are cofounders of ACE NanoMed Ltd. K.W.L. received research funding from Viracta Therapeutics and ScinnoHub Pharmaceutical Co., Ltd. B.Y.Y.M. is a consultant for Viracta Therapeutics, Alentis and Y-biologics, serves on the advisory boards of MSD, BMS, Novartis and Taiho; and has received research funding from Boerhinger Ingelheim and Merck Serono. All other authors declare no competing interests. J.Y.K.C. is cofounder of Agilis Robotics and has received research funding from MSD.

## Additional information

[1]Department of Anatomical and Cellular Pathology, Prince of Wales Hospital, The Chinese University of Hong Kong, Hong Kong SAR, China. [2]State Key Laboratory of Translational Oncology, Sir YK Pao Centre for Cancer, The Chinese University of Hong Kong, Hong Kong SAR, China. [3]Ming Wai Lau Centre for Reparative Medicine, Karolinska Institutet, Shatin, Hong Kong SAR, China. [4]Department of Surgery, Prince of Wales Hospital, The Chinese University of Hong Kong, Hong Kong SAR, China. [5]Center for Neuromusculoskeletal Restorative Medicine, Hong Kong Science Park, Hong Kong SAR, China. [6]College of Chemistry and Green Catalysis Center, Zhengzhou University, Zhengzhou, China. [7]The Jackson Laboratory for Genomic Medicine, Farmington, CT 06032, USA. [8]School of Biological and Health Systems Engineering, Arizona State University, Tempe, AZ 85281, USA. [9]Department of Otorhinolaryngology, Head and Neck Surgery, Prince of Wales Hospital, The Chinese University of Hong Kong, Hong Kong SAR, China. [10]Department of Clinical Oncology, State Key Laboratory of Translational Oncology, Charlie Lee Precision Immuno-oncology program, Sir Y.K. Pao Centre for Cancer, The Chinese University of Hong Kong,

Hong Kong SAR, China. [11]Graduate Institute and Department of Microbiology, College of Medicine, National Taiwan University, Taipei 100233, Taiwan. [12]Department of Genetics and Genome Sciences, University of Connecticut Health Center, Farmington, CT 06030, USA. [13]Institute for Systems Genomics, University of Connecticut Health Center, Farmington, CT 06030, USA. [14]The Jackson Laboratory Cancer Center, Bar Harbor, ME 04609, USA. [15]State Key Laboratory of Stem Cell and Reproductive Biology, Institute of Zoology, Chinese Academy of Sciences, Beijing, China. [16]Key Laboratory of Organ Regeneration and Reconstruction, Chinese Academy of Sciences, Beijing, China. [17]Beijing Institute for Stem Cell and Regenerative Medicine, Beijing, China. [18]University of Chinese Academy of Sciences, Beijing, China. [19]These authors contributed equally: Man Wu, Pok Man Hau, Linxian Li. [20]These authors jointly supervised this work: Albert Wu Cheng, Kwok-Wai Lo. ✉e-mail: albert.cheng@ioz.ac.cn; kwlo@cuhk.edu.hk

