## [Peer Review File · Nature Communications]

REVIEWER COMMENTS

Reviewer #1 (Remarks to the Author):

The paper by Wu et al. describes multiple experimental systems that have the ability to induce expression of BZLF1 through transactivation of its promoter in EBV-associated carcinomas, with a focus on nasopharyngeal carcinoma. Here BZLF1 expression launched the EBV lytic cascade and caused tumor cell death, both in vitro and in animal models. This is a highly technical paper that would benefit from a more extensive information about the used experimental systems that will not be necessarily known to EBV researchers. The paper reports the use of three experimental systems to activate BZLF1 expression: one based on the CRISPR-Casilio system, one based on a TALE activator, on one based on a mRNA that encodes this TALE activator. The link between the CRISPR-Casilio system and the TALE system is tenuous and appears artificial. Moreover, only the mRNA-based system has the potential of being tested one day in a clinical setting. I would have preferred to see a single experimental system investigated more thoroughly. The results of the mRNA-based treatment of NPC in an animal model in vivo are particularly interesting, although the evidence that the observed effect is the result of cell destruction caused by lytic replication only is lacking. It would be important to gather more information on the mechanism that caused cell death. Generally, although many of the experiments I suggested were limited to the mRNA-based experiments, they would also be relevant to the CRISPR-Casilio system.

Major comments

- 1) Fig.1 More information is needed about the experimental system used. What are the viruses described line 135? It would be important to determine whether the treated cells complete the induced lytic replication, the BGLF4 western blot suggests that this is not the case for C666-1 and C17. Thus, immunostains (IF, western blot) of the treated cells for gp350 are necessary.
- 2) Fig.3b,c The efficacy of BZLF1 induction is impressive. However, the expression of the downstream members of the lytic cascade seem to be less efficient. It would be important to perform IF stain for EA and/or BGLF4, but also for gp350 to determine the percentage of cells that have progressed through and successfully completed the EBV lytic cycle.
- 3) Fig. 5 Here again it would be important to know whether late proteins such as gp350 are expressed in the tumor cells. Analysis at 24 hours post injection of the LNA might be too early for late protein expression, but the authors have tumor samples at a later time point as in Suppl. Fig. 3. The BZLF1 staining in b) appears to be mainly intracytoplasmic (compare with a)). How can that be? We know that the active form of BZLF1 is mainly nuclear. The authors need to quantify the percentage of positive cells in the investigated sections.

4) Fig.6. It would be very important to perform BZLF1, EA, BGLF4 and gp350 stains on the tumors generated with Xeno76 PDX as was performed in Fig.5 for the cell lines.

5) Fig.6 The authors need to characterize the inflammatory infiltrate around the tumor cells, are these T or B lymphocytes? NK cells? Are there macrophages visible?

6) Fig.6b The authors need to provide a statistical analysis of the results, including a comparison between cells treated with the LNP alone or in combination with ganciclovir.

7) I am surprised that the authors did not investigate more gastric carcinoma and NPC cell lines. At the very least they should test NPC43 as they did in Fig. 3 for the other cell lines.

8) The authors should state more clearly and quantify precisely the benefit, if any, of the combined LNP ganciclovir therapy relative to LNP alone.

9) The authors need to disclose the sequence of the TALE and of the mRNA used in the paper.

Minor comments: there are multiple typos throughout the text that need to be corrected (e.g. line 109, 133, 256)

Reviewer #2 (Remarks to the Author):

The present work describes the development of CRISPR-Cas9-based and TALE-based transcriptional activators that specifically activate the endogenous BZLF1 gene and efficiently induce downstream lytic genes in EBV-positive cancers. The CRISPR system was first used to screen for candidate genomic sites accessible for binding, and the TALE system was employed to target the optimal binding site identified by the CRISPR system for artificial activation of BZLF1 and lytic genes. Overall, the study is comprehensive, encompassing a significant amount of experimental data obtained from both cell cultures and animal models. However, there is insufficient justification provided for using the CRISPR system, as the proposed TALE system may also be used to screen for candidate genomic sites. Moreover, the authors stated that "it is challenging to effectively deliver all the three complex and large constructs" (lines 164-165) pertaining to the CRISPR system. In other words, the use of the CRISPR system does not appear to add

any scientific value to the current study. Additionally, the authors claimed that "no obvious off-target effect of TZ3 was observed on EBV-negative human cells" (lines 183-184). To support this claim, the authors should provide experimental evidence demonstrating that TZ3 indeed does not elicit any off-target effects.

Reviewer #3 (Remarks to the Author):

In this manuscript, Wu et al developed the CRISPR-Casilio and TALE activator to activate the expression of the BZLF1 gene that can reactivate the lytic cycle and induce cytolytic effect in EBV-positive epithelial cancers. The nucleoside-modified mRNA encoding TALE lytic activator was delivered by LNP into cells and in vivo, and the therapeutic efficacy was demonstrated. Overall, the experimental designs were logical. The conclusions were partly supported by the observed experimental results and more descriptions for some experiments should be provided to help the reader understand the research. Several concerns as shown below need to be addressed before it's suitable for publication.

1. In line 144, when testing the the CRISPR-Casilio system, the authors stated that "Treatment of stably transfected cells with doxycycline (Dox) resulted in transactivator induction and the abundant expression of Zta, Rta, and downstream lytic proteins (BGLF4 and VCA) in EBV-positive epithelial cancer cells". However, in the data shown in Fig 1d, among 2 out of the 3 cell lines tested, the BGLF4 and VCA were barely expressed or not expressed at all based on western blotting. In Supplementary Fig 2a, based on qPCR, BGLF4 also was not expressed in C17 cells. Therefore, the statement was not supported by the observed results. For therapeutic purposes, can the authors comment on whether this a concern? Although these downstream lytic proteins were not induced at all in C17 cells, the inhibition of cell viability and colony formation seemed to be as effective as in other 2 cell lines. Was it due to that other lytic proteins were induced in C17 cells to achieve this effect and what other lytic proteins? Was the induction of downstream genes depending on the cell lines and will it lead to inconsistency of the lytic effects? Similar results were observed when testing TALE based activator in C666-1 cells in Fig 2b.

2. In Supplementary Fig 2a, what is the role of triton X-100 detergent? Does it help or disrupt the LNP formation? From the results, it looked like it's required for the LNP formation. Maybe the authors should be described to help readers not in the LNP delivery field understand.

3. The order of supplementary figures should be arranged following the order of description in the text – Fig S3c was actually discussed after Fig S4 and S5. Also, in FigS3c, would the high uptake of LNP-encapsulated mRNA in the liver be a concern?

4. In Fig 5, what is the significance of EA-D and BMRF1? Why checking these proteins? It should be explained in the text.

5. There are several typos and grammatical errors, the authors should carefully proofread. For example:

Line 98: ""board-spectrum

Line 113: activation "of" and proven the usefulness

Line 217: Fig 2e should be Fig 2f

Line 295: could not be effectively "induce" when

Reviewer #4 (Remarks to the Author):

In this manuscript, authors reported an mRNA therapeutic approach for targeted therapy of EBV-positive cancers. Authors designed an mRNA encoding the BZLF1-specific transcriptional activator (mTZ3) and prepared lipid nanoparticles (LNPs) to encapsulate the mTZ3 for systemic delivery (mTZ3-LNPs). Authors demonstrated that cellular treatment of mTZ3-LNPs activated EBV lytic gene expression and intravenous injection of mTZ3-LNPs induced selective anti-tumor effects in EBV-positive tumors via mRNA-based lytic induction. Overall, this manuscript is well written and showed solid results to support the conclusions. However, I would like to suggest that authors consider following comments to improve the manuscript before publication.

1. In figure 5, authors only showed the expression of EBV lytic genes in the tumor tissue sections after mTZ3-LNP injection. If authors provide data on the cellular distribution of LNPs in the tumor tissues, it will help readers understand how mTZ3-LNPs induce the anti-tumor effects. And cellular expression of EBV lytic genes should be also studied by flow cytometry.

2. To demonstrate the clinical translatability of mTZ3-LNPs, it is necessary to compare the therapeutic efficacy of mTZ3-LNPs (+GCV) with conventional treatments for EBV-positive cancers, particularly in in vivo anti-tumor studies.

3. Unlike the schematic in figure 7, co-treatment with antiviral ganciclovir did not augment the anti-tumor effect of mRNA-loaded LNPs in EBV-positive tumors. Why? Authors should explain this in Discussion section.

4. As shows in supplementary figure 3, most of the injected LNPs accumulated in the liver. To support the safety of mTZ3-LNPs, authors should investigate whether intravenous injection of mTZ3-LNPs induces the expression of EBV lytic genes in the liver tissues.

Reviewer #5 (Remarks to the Author):

The authors show an extensive work on the potential application of a lipid nanoparticle loaded with a mRNA in the EBV-associated cancers. The data confirmed the efficiency of the nanosystem in vitro and in vivo. However, some additional information could be provided:

- 1) The nanoparticles could be characterized by dynamic light scattering in order to confirm that the nanoparticles were formed as expected and to know the size distribution including the polydispersity index.
- 2) Does the nanoparticle pass through cell uptake process? Confocal microscopy of the cells incubated with fluorescent nanoparticles could prove it.
- 3) Please include additional information about LSM measurements. Include equipment details and wavelength of excitation/emission.
- 4) The in vivo assays were performed for 2 weeks only. The authors could include justification for this choice.
- 5) How long is the circulation time of the nanoparticles? The blood was analyzed by fluorescence?

Point-by-point reply:

Thank you so much for passing on the reviewers' comments and suggestions received so far and for the opportunity to revise our manuscript NCOMMS-23-24266 entitled "Synthetic *BZLF1*-targeted transcriptional activator for efficient lytic induction therapy against EBV-associated epithelial cancers", according to the helpful comments from the reviewers. We are grateful to the reviewers' expert comments to further improve the manuscript in terms of clarity and impact. We have, therefore, revised the manuscript addressing all the comments of the reviewers in the text, as well as listed as "point-by-point reply" below for your kind consideration.

In the revised manuscript and our response to the reviewers' comments, we have clearly explained our strategy for development of synthetic transcriptional activator for lytic induction therapy in patients with EBV-associated epithelial cancers. As suggested, we sought and included multiple new EBV-positive cancer cell lines (YCCEL1, NPC43-M81, AGS-EBV and Akata-EBV) in our study to demonstrate the high efficiency of mTZ3-LNPs in inducing EBV lytic gene expression. We conducted comprehensive *in vitro* and *in vivo* analysis of the lytic cascade, including the expression of immediate early lytic genes/proteins (*BZLF1/Zta*, *BRLF1/Rta*), early lytic genes/ proteins (*BMRF1/EA-D*, *BGLF4*) and late lytic genes/proteins (VCA and *BLLF1/gp350*) using western blotting, flow cytometry analysis, immunofluorescence and immunohistochemical staining, and RNAscope RNA *in-situ* hybridization. For the *in vivo* study, the inflammatory infiltrates around xenograft tumor cells in NOD-SCID mouse models were examined. In the introduction and discussion sections of our revised manuscript, we have added a detailed discussion of the cytotoxic effects of early and late lytic proteins during lytic reactivation and the reasons for the including GCV in EBV lytic induction therapy.

Reviewer #1:

Overall comments:

The paper by Wu et al. describes multiple experimental systems that have the ability to induce expression of *BZLF1* through transactivation of its promoter in EBV-associated carcinomas, with a focus on nasopharyngeal carcinoma. Here *BZLF1* expression launched the EBV lytic cascade and caused tumor cell death, both *in vitro* and in animal models. This is a highly technical paper that would benefit from a more extensive information about the used experimental systems that will not be necessarily known to EBV researchers. The paper reports the use of three experimental systems to activate *BZLF1* expression: one based on the CRISPR-Casilio system, one based on a TALE activator, on one based on a mRNA that encodes this TALE activator. The link between the CRISPR-Casilio system and the TALE system is tenuous and appears artificial. Moreover, only the mRNA-based system has the potential of being tested one day in a clinical setting. I would have preferred to see a single experimental system investigated more thoroughly. The results of the mRNA-based treatment of NPC in an animal model *in vivo* are particularly interesting, although the evidence that the observed effect is the result of cell destruction caused by lytic replication only is lacking. It would be important to gather more information on the mechanism

that caused cell death. Generally, although many of the experiments I suggested were limited to the mRNA-based experiments, they would also be relevant to the CRISPR-Casilio system.

Our Reply:

Thank you so much for your suggestions and comments. In this study, we aimed to exploit a novel strategy to develop an efficient lytic induction therapy against EBV-associated epithelial cancers, thus bypassing cellular restrictions on the EBV latent-lytic switch. As shown in our original manuscripts, both CRISPR-Casilio-based and TALE-based transcriptional activators were designed and constructed to artificially activate the EBV immediate early lytic gene *BZLF1*, subsequently inducing lytic reactivation in EBV-positive gastric cancer (SNU719) and NPC (C666-1) cell lines. Although both systems are amenable to bioengineering to enable binding to specific sequences in *BZLF1* promoter, sgRNAs from the CRISPR-Casilio activator system are convenient to synthesis and use for screening and identifying target sequences for efficient transcriptional activation. Thus, we first used the CRISPR-Casilio activator system and multiple sgRNAs (sgRNA1, sgRNA2, sgRNA3, sgRNA4) to identify the target region in the *BZLF1* promoter with the highest efficiency for transcriptional activation. We demonstrated that the EBV-positive tumor cells treated with the CRISPR-Casilio activator system and sgRNA3 expressed the highest levels of BZLF1/Zta and its downstream lytic proteins. Furthermore, as shown in Supplementary Figure 2, the establishment of an inducible Casilio (iCasilio) activator system allowed us to clearly demonstrate the potent effect of synthetic *BZLF1*-specific transcriptional activator in terms of activating EBV lytic cycle and killing tumor cells in both *in vitro* and *in vivo* EBV-positive cancer models. The findings provide evidence to support our effort to develop artificial transcriptional factor that can induce EBV lytic reactivation as an efficient therapeutic strategy against EBV-associated epithelial cancers. Although the complexity of the system may hinder the effectiveness of the CRISPR-Casilio-based activator to the patient, our study has delineated the potential activator-binding regions in *BZLF1* promoter.

In contrast to the CRISPR-Casilio-based activator, the TALE-based activator system was selected for the development of an efficient lytic induction therapy because of its feasibility for *in vivo* therapeutic delivery. As the construction of sequence-specific TALE plasmids involves multiple cloning steps and validation processes, we then designed the TALE construct TZ3 based on the sgRNA3-binding sequence identified from our successful CRISPR-Casilio activator system to induce EBV reactivation. To determine the concordance between the two systems, we also constructed additional TALE constructs (TZ1, TZ2, and T4) that targeted the corresponding binding sequences of other sgRNAs identified in this study. The consistent findings of the two systems support the usefulness of the CRISPR-Casilio-based system for the comprehensive screening of the effective transcriptional activator-binding sites.

After proving the high efficiency and specificity of our TALE-based activator system for the reactivation of endogenous *BZLF1* in EBV-positive cancer cells, we further applied state-of-art nucleoside-modified mRNA and lipid nanoparticle (LNP) technologies to deliver an artificial *BZLF1*-specific transcriptional activator to animal models in *in vivo* experiments. The current study thus demonstrates a comprehensive strategy for the development of mRNA nanomedicines and synthetic transcriptional factors to activate potential therapeutic targets in human cancers.

On the other hand, we further demonstrated that the lytic genes induced by mTZ3-LNP in EBV-positive epithelial cells promoted cell death, even when only abortive lytic cycle was reactivated as in C666-1 cells. Studies have shown that the induction of both the early and late phases of EBV lytic cycle contribute to the death of EBV-positive epithelial cancer cells. In addition to cell destruction by EBV virions, the overexpression of immediately early and early lytic proteins was shown to be involved in the killing of NPC and gastric cancer cells (Hui et al., 2012). The expression of Zta was shown to induce G2/M arrest and p53 phosphorylation in tumor cells (Mauser et al, 2002). Studies have reported that Rta irreversibly induces an G1 arrest and cellular senescence in NPC cells (Feng et al. 2002; Wang et al. 2008; Chen et al, 2011). Other early viral proteins, such as the EBV alkaline exonuclease, BGLF5, could switch off most cellular gene functions and trigger apoptotic machinery (Zuo et al, 2008). Notably, in Figures 3d and 4d of revised manuscript, we show that cleaved caspase 3 was detected in the mTZ3-LNP-treated EBV-positive epithelial cells, either with induction of late lytic proteins (VCAp18 and Gp350) (SNU719, YCCEL1) or with reactivation of abortive lytic cycle (C666-1, NPC43-M81).

References:

- Hui KF, Ho DN, Tsang CM, Middeldorp JM, Tsao GS, Chiang AK. Activation of lytic cycle of Epstein-Barr virus by suberoylanilide hydroxamic acid leads to apoptosis and tumor growth suppression of nasopharyngeal carcinoma. *Int J Cancer*. 2012;131(8):1930-40.
- Mauser A, Saito S, Appella E, Anderson CW, Seaman WT, Kenney S. The Epstein-Barr virus immediate-early protein BZLF1 regulates p53 function through multiple mechanisms. *J Virol*. 2002; 76: 12503-12.
- Mauser A, Holley-Guthrie E, Simpson D, Kaufmann W, Kenney S. The Epstein-Barr virus immediate-early protein BZLF1 induces both a G(2) and a mitotic block. *J Virol*. 2002; 76: 10030-7.
- Feng WH, Westphal E, Mauser A, Raab-Traub N, Gulley ML, Busson P, Kenney SC. Use of adenovirus vectors expressing Epstein-Barr virus (EBV) immediate-early protein BZLF1 or BRLF1 to treat EBV-positive tumors. *J Virol*. 2002; 76: 10951-9.
- Wang L, Shan L, Lo KW, Yin J, Zhang Y, Sun R, Zhong J. Inhibition of nasopharyngeal carcinoma growth by RTA-expressing baculovirus vectors containing oriP. *J Gene Med* 2008; 10: 1124-33.
- Chen YJ, Tsai WH, Chen YL, Ko YC, Chou SP, Chen JY, Lin SF. Epstein-Barr Virus (EBV) Rta-mediated EBV and Kaposi's sarcoma-associated herpesvirus lytic reactivations in 293 cells. *PLoS One*, 2011; 6: e17809.
- Zuo J, Thomas W, van Leeuwen D, Middeldorp JM, Wiertz EJ, Rensing ME, Rowe M. The DNase of gammaherpesviruses impairs recognition by virus-specific CD8+ T cells through an additional host shutoff function. *J Virol*. 2008; 82: 2385-93.

Major comments:

- 1) Fig.1 More information is needed about the experimental system used. What are the viruses described line 135? It would be important to determine whether the treated cells complete the

induced lytic replication, the BGLF4 western blot suggests that this is not the case for C666-1 and C17. Thus, immunostains (IF, western blot) of the treated cells for gp350 are necessary.

Our Reply:

In the revised manuscript, we added detailed information and additional references pertaining to the CRISPR-Casilio- and TALE-based systems used to artificially induce endogenous *BZLF1* expression in EBV-positive cancer cells to the Introduction, Results and Method sections and Supplementary Materials. In line 135 of the original manuscript, the “viruses” was used to represent the “lentiviral vectors” encoding (1) HA-dCas9-EGFP; (2) 3xFLAG136 PUFa-p65HSF and; (3) a panel of sgRNAs-5xPBSa. We have replaced the term “viruses expressing ...” with “lentiviral vectors encoding...” in the revised manuscript.

Of the three EBV-positive cancer cell lines engineered with doxycycline (Dox)-inducible Casilio activators and sgRNAs targeting *BZLF1*, we showed that the DOX-treated SNU719 and C17 cells expressed immediately early (Zta, Rta), early (BGLF4) and late lytic proteins (VCAp18, gp350) in the revised Figures 1d and 1e, and Supplementary Figure S2a. As shown in Supplementary Figure S2c, the completed lytic cycle induced in the SNU719 and C17 cell lines was also demonstrated by the production of infective EBV virions after induction of the CRISPR-Casilio-based activator by DOX treatment (Supplementary Figure 2b). In C17 cells treated with Dox for 96 h, the expression of proteins encoded by all the above-mentioned lytic genes was demonstrated by western blotting (Figure 1d). The results confirmed that the treated cells could complete the induced lytic replication. Distinct from the SNU719 and C666-1 with high copy numbers of EBV episomes (~20-50), C17 contain only 2-3 copies of virus genomes per cell. (Nakayama, 2019; Yip. 2018). Accordingly, C17 cells with low copy numbers of EBV genomes require a longer period to accumulate sufficient levels of Zta and Rta proteins for the transcriptional activation of downstream early and late lytic genes

Because of the abortive nature of EBV lytic reactivation in C666-1, late lytic proteins (VCAp18 and BLLF1/gp350) were not detected in this cell line. In our previous studies, we demonstrated that only an abortive lytic cycle could be induced in C666-1 cells which harbor a nonsense mutation in an EBV lytic gene, *BNRF1* (Hui, et al, 2012; Tso et al, 2013). This finding suggests the absence of a functional BNRF1 lytic protein, which would be required to maintain late gene expression in C666-1 cells. In the absence of BNRF1, the formation of a nuclear membrane-less replication compartment for EBV lytic cycle amplification, the expression of late lytic proteins and the production of virions were disrupted (Yiu et al, 2022). In our experiments, we confirmed the induction of immediately early (BZLF1, BRLF1) and early (BGLF4) lytic protein expression in Dox-treated C666-1 cells. Nevertheless, abortive EBV lytic cycle also led to cell death in C666-1 via apoptosis induction, as reported previously (Hui, et al., 2012). The induction of an abortive lytic cycle was also found in the NPC43-M81 cell line (Figure 4d).

In addition to the late lytic protein VCAp18, we included another late lytic protein BLLF1/gp350 in our IF, IHC and flow cytometry experiments in the revised manuscript as suggested. (Figures 1e, 3e, 3f and 6a-d and Supplementary Figure 9)

References:

Hui KF, Ho DN, Tsang CM, Middeldorp JM, Tsao GS, Chiang AK. Activation of lytic cycle of Epstein-Barr virus by suberoylanilide hydroxamic acid leads to apoptosis and tumor growth suppression of nasopharyngeal carcinoma. *Int J Cancer*. 2012 Oct 15;131(8):1930-40.

Tso KK, Yip KY, Mak CK, Chung GT, Lee SD, Cheung ST, To KF, Lo KW. Complete genomic sequence of Epstein-Barr virus in nasopharyngeal carcinoma cell line C666-1. *Infect Agent Cancer*. 2013 Aug 2;8(1):29.

Yiu SPT, Guo R, Zerbe C, Weekes MP, Gewurz BE. Epstein-Barr virus BNRF1 destabilizes SMC5/6 cohesin complexes to evade its restriction of replication compartments. *Cell Rep*. 2022 Mar 8;38(10):110411.

Nakayama A, Abe H, Kunita A, Saito R, Kanda T, Yamashita H, Seto Y, Ishikawa S, Fukayama M. Viral loads correlate with upregulation of PD-L1 and worse patient prognosis in Epstein-Barr Virus-associated gastric carcinoma. *PLoS One*. 2019 Jan 29;14(1):e0211358.

Yip, Y.L., et al. Establishment of a nasopharyngeal carcinoma cell line capable of undergoing lytic Epstein-Barr virus reactivation. *Lab Invest*. 98, 1093-104 (2018).

2) Fig.3b,c The efficacy of BZLF1 induction is impressive. However, the expression of the downstream members of the lytic cascade seem to be less efficient. It would be important to perform IF stain for EA and/or BGLF4, but also for gp350 to determine the percentage of cells that have progressed through and successfully completed the EBV lytic cycle.

Our Reply:

In the revised manuscript, representative immunofluorescence staining images to detect Zta, EA-D and gp350 expression in SNU719 and C666-1 cells treated with mTZ3-LNPs for 96 h are shown in Figure 3e. The expression of Zta, Rta and early (BGLF4, BMRF1/EA-D) and late (VCAP18) lytic proteins in EBV-positive cancer cells treated with mTZ3-LNPs for 3 to 96 h were also determined by Western blotting (Figure 3d). Our findings indicated that Zta was induced in SNU719 and C666-1 cells at 12 and 18 h post-mTZ3-LNP treatment. The early lytic proteins BGLF4 and BMRF1/EA-D were expressed in SNU719 and C666-1 cells after 18 and 48 h of mTZ3-LNP treatment, respectively. We also detected the late lytic proteins VCAP18 and BLLF1/gp350 in SNU719 cells. In C666-1 cells treated with mTZ3-LNP, only immediately early (Zta, Rta) and early (BGLF4 and EA-D) proteins were detected. As discussed earlier, only an abortive EBV lytic cycle can be induced in C666-1 cells. Although EBV late lytic proteins (VCAP18 and gp350) were not expressed in C666-1 cells after mTZ3-LNP treatment, apoptosis induction, indicated by cleaved caspase 3, was observed after the reactivation of abortive lytic cycle as reported previously (Hui *et al.* 2012). The percentages of SNU719 cells that have progressed through the EBV lytic cycle were also determined in a flow cytometry analysis using antibodies specific for Zta, EA-D and gp350 (Figure 3f). gp350 expression was detected in 20.1% and 26.4% of SNU719 cells treated with mTZ3-LNPs for 72 and 96 h, respectively.

Reference:

Hui KF, Ho DN, Tsang CM, Middeldorp JM, Tsao GS, Chiang AK. Activation of lytic cycle of Epstein-Barr virus by suberoylanilide hydroxamic acid leads to apoptosis and tumor growth suppression of nasopharyngeal carcinoma. *Int J Cancer*. 2012 Oct 15;131(8):1930-40.

3) Fig. 5 Here again it would be important to know whether late proteins such as gp350 are expressed in the tumor cells. Analysis at 24 hours post injection of the LNA might be too early for late protein expression, but the authors have tumor samples at a later time point as in Suppl. Fig. 3. The BZLF1 staining in b) appears to be mainly intracytoplasmic (compare with a)). How can that be? We know that the active form of BZLF1 is mainly nuclear. The authors need to quantify the percentage of positive cells in the investigated sections.

Our Reply:

As suggested, we performed immunohistochemical staining to detect the expression of the late protein BLLF1/gp350 in the SNU719 tumor cells at 12, 24 and 48 hours post-mTZ3-LNP treatment (Figure 5 of original version). Representative images are shown in Figure 6a of revised manuscript. The percentages of tumor cells expressing the lytic proteins Zta, EA-D and gp350 in the investigated sections were determined using ImageJ program. The findings are shown in Figure 6b of our revised manuscript.

The original figure 5b depicted our detection of *BZLF1*, *BMRF1* and *BGLF4* transcripts in the tumors using RNAscope RNA *in-situ* hybridization. The staining signals in this figure were the *BZLF1*, *BMRF1* and *BGLF4* mRNAs induced by the synthetic TZ3 transcriptional activator. The mRNA signals within the cells expressing *BZLF1* transcripts were mainly intracytoplasmic (red arrows). In the cells with abundant *BZLF1*, *BMRF1* or *BGLF4* transcripts, the signals fused together and occupied both the nuclear and cytoplasmic areas (blue arrow). In the new figure 6c of revised manuscript, the expression of *BZLF1*, *BGLF4*, *BMRF1* and *BLLF1* mRNA detected by RNAscope RNA *in-situ* hybridization are shown. The percentages of tumor cells expressing immediate early (*BZLF1*), early (*BGLF4*, *BMRF1*) and late (*BLLF1*) lytic gene transcripts were determined using the ImageJ program and are shown in Figure 6d of the revised manuscript.

4) Fig.6. It would be very important to perform BZLF1, EA, BGLF4 and gp350 stains on the tumors generated with Xeno76 PDX as was performed in Fig.5 for the cell lines.

Our Reply:

In Figure 6 of the original manuscript, the Xeno76 PDX tumors in mice treated with multiple intravenous injections of vehicle, GCV alone or mTZ3-LNP alone or in combination with GCV were also subjected to immunohistochemical staining of BZLF1/Zta, EA-D and gp350. Expression of the lytic proteins Zta, EA-D and gp350 was detected in the residual tumors harvested from mice treated with mTZ3-LNP, but not those treated with vehicle and GCV controls. We detected few EA-D- and gp350-positive tumor cells in mice treated with combined mTZ3-LNP

and GCV, probably due to the cytotoxic effects of BGLF4-converted GCV in the lytic-reactivated tumor cells. Representative images are shown in Supplementary Figure 9 of the revised manuscript.

5) Fig.6 The authors need to characterize the inflammatory infiltrate around the tumor cells, are these T or B lymphocytes? NK cells? Are there macrophages visible?

Our Reply:

Per your suggestion, we characterized the inflammatory infiltrate around the tumor cells by detecting markers of mouse B cells (CD19), T cells (CD8A), macrophages (CD68) and NK cells (NKP46). As NOD-SCID mice do not produce T and B lymphocytes, neither CD19-positive nor CD8A-positive cells were found in the tumors or in the normal thymus tissues of the mice. As shown in Supplementary Figure 12, abundant NK cells were observed in the residual tumors in the mice treated with mTZ3-LNP alone or in combination with GCV. Macrophages were also detected in the necrotic lesions and residual tumors (arrows) in mice treated with mTZ3-LNP or in combination with GCV. Nevertheless, NOD-SCID mouse models of EBV-associated cancers are not suitable for investigating the effects of lytic induction therapy on innate and adaptive immune response induction given the lack of B and T lymphocytes. We have discussed these limitations in the Discussion section of our revised manuscript. Although we have presented our preliminary findings regarding the occurrence of NK cells and macrophages in the tumors treated with mTZ3-LNP with and without GCV and control treatments in Supplementary Figure 12, the effects of EBV lytic induction therapy on the induction of innate and adaptive immune response against the tumor must be proved in an immunocompetent model. In the Discussion section of our revised manuscript, we have proposed to further characterize the immune responses following mTZ3-LNP-based lytic induction therapy and the synergistic antitumor effects of this treatment and anti-PD1/PDL1 therapy in EBV-positive tumor xenografts in humanized mouse models.

6) Fig.6b. The authors need to provide a statistical analysis of the results, including a comparison between cells treated with the LNP alone or in combination with ganciclovir.

Our Reply:

In Figure 7b and 7c of the revised version (Fig. 6b of original manuscript), we have included the results of statistical analysis of our *in vivo* experiments to compare the antitumor effects of mTZ3-LNP alone, either alone or in combination with GCV, with those of control treatments in the EBV-positive epithelial cancers. We have also included a comparison of the groups treated with mTZ3-LNP alone or in combination with GCV. While potent growth inhibitory effects were observed in the tumors treated with mTZ3-LNP alone or in combination with GCV, no significant difference in the tumor volume/tumor index was observed between these two treatment groups. As mentioned in the Discussion section, no obvious bystander killing effect was observed in the combined mTZ3-LNP+GCV treatment group because of the high efficiency of lytic induction and potent cytotoxicity of mTZ3-LNP in EBV-positive tumor cells.

7) I am surprised that the authors did not investigate more gastric carcinoma and NPC cell lines. At the very least they should test NPC43 as they did in Fig. 3 for the other cell lines.

Our Reply:

According to the reviewer's comments, we further investigated the effect of mTZ3-LNP on the reactivation of Zta expression in additional EBV-positive cancer cell lines, including native EBV-infected (YCCEL1, Akata-EBV+) and EBV re-infected (NPC43-M81, AGS-EBV) cancer cells (Figure 4). Some EBVaGC cell lines (e.g. NCC24) were not included in our study because only ~10-20% of cells in that cell lines are EBV-positive. Those cell lines with a heterogenous EBV infection status is not suitable for experiments to determine the effect of our mTZ3-LNP on EBV lytic induction. In the NPC43 cell line, only around 40-60% of the cells are EBV-positive. Thus, we preferred to conduct our experiments in NPC43-M81 cells, which have a homogenous EBV infection status, to investigate the efficiency of mTZ3-LNP in inducing the EBV lytic reaction (Figure 4a and 4d). NPC43-M81 was established by infection of EBV-negative NPC43 cells with M81 strain recombinant EBV. The EBV-positive rate among NPC43-M81 cells is maintained at approximately 100%. As we demonstrated in Figure 3 of the original manuscript, we assessed the effect of mTZ3-LNP treatment on induction of Zta, Rta and downstream early (BGLF4, EA-D) and late (VCAP18) lytic protein expression in YEECL1 and NPC43-M81 cells by Western blotting (Figure 5d). Furthermore, we investigated the *in vivo* antitumor effect of mTZ3-LNP treatments in C17 cell-derived xenograft models (Fig 7). The synthetic BZLF1-targeted transcriptional activator efficiently suppressed the growth of EBV-positive epithelial tumors in NOD-SCID mouse models, even those with a low copy number of EBV episomes (e.g., C17).

8) The authors should state more clearly and quantify precisely the benefit, if any, of the combined LNP ganciclovir therapy relative to LNP alone.

Our Reply:

Ganciclovir is an essential component in the clinical trials of EBV lytic induction therapies and oncolytic virus therapies. We have further described the importance of including ganciclovir (GCV) in EBV lytic induction therapy in the Introduction and Discussion sections, as follows:

“In this type of lytic induction therapy, antiviral ganciclovir (GCV) is co-administrated with the chemical inducer to patients to mediate specific cell killing and prevent virus production in EBV-infected cells. GCV is non-cytotoxic to EBV-positive tumors with restricted viral latency. However, BGLF4, an EBV-encoded serine/threonine kinase is expressed during lytic reactivation and can convert the non-cytotoxic GCV to an active, cytotoxic form via phosphorylation. This BGLF4-converted cytotoxic GCV or phosphorylated GCV rapidly kills cancer cells and has been evaluated in clinical trials of oncolytic therapy against EBV-associated cancers (3, 7-9). In addition to mediating the direct killing of EBV-positive tumor cells after lytic reactivation, the phosphorylated GCV can be transferred to adjacent cells, leading to a “bystander killing” effect.

Importantly, phosphorylated GCV can inhibit EBV encoded DNA polymerase, interrupting the production of infective virions and preventing dissemination of virus during lytic induction therapy (7-11).”

“Through in vivo study, we observed no significant difference in growth inhibition between tumors treated with mTZ3-LNP alone or in combination with GCV. The absence of an obvious bystander effect with combined treatment may be due to the high efficiency of lytic reactivation and the potent cytotoxic effect of mTZ3-LNP on EBV-positive tumors. Despite of the limited bystander killing effect observed, the GCV administration is an essential procedure for rapid and specific killing of the EBV-positive cells and inhibit the production of infective virions during EBV lytic induction treatment.”

9) The authors need to disclose the sequence of the TALE and of the mRNA used in the paper.

Our Reply:

The amino acid sequences of the TALEs and the sequence of the mTZ3 transcript are listed in Supplementary Table 3 of our revised manuscript.

Minor comments: there are multiple typos throughout the text that need to be corrected

(e.g. line 109, 133, 256)

Our Reply:

Our manuscript was checked for English language usage by a professional editor, and typos have been corrected in the revised manuscript.

Reviewer #2:

Overall comments:

The present work describes the development of CRISPR-Cas9-based and TALE-based transcriptional activators that specifically activate the endogenous BZLF1 gene and efficiently induce downstream lytic genes in EBV-positive cancers. The CRISPR system was first used to screen for candidate genomic sites accessible for binding, and the TALE system was employed to target the optimal binding site identified by the CRISPR system for artificial activation of BZLF1 and lytic genes. Overall, the study is comprehensive, encompassing a significant amount of experimental data obtained from both cell cultures and animal models. However, there is insufficient justification provided for using the CRISPR system, as the proposed TALE system may also be used to screen for candidate genomic sites. Moreover, the authors stated that “it is

challenging to effectively deliver all the three complex and large constructs" (lines 164-165) pertaining to the CRISPR system. In other words, the use of the CRISPR system does not appear to add any scientific value to the current study. Additionally, the authors claimed that "no obvious off-target effect of TZ3 was observed on EBV-negative human cells" (lines 183-184). To support this claim, the authors should provide experimental evidence demonstrating that TZ3 indeed does not elicit any off-target effects.

Our reply:

We thank the reviewer very much for the comments. In the revised manuscript, we have clearly described the scenario for using the CRISPR-Casilio-based and TALE-based systems to develop an effective artificial transcriptional activator for inducing lytic reactivation in EBV-positive cancer cells. Although both systems are amenable to bioengineering to enable the binding of specific sequences in the *BZLF1* promoter, the sgRNAs used in the CRISPR-Casilio activator system are convenient to synthesize and use for screening and identifying target sequences for efficient transcriptional activation. Thus, we first used the CRISPR-Casilio activator system and multiple sgRNAs (sgRNA1, sgRNA2, sgRNA3, sgRNA4) to identify the target region in the *BZLF1* promoter with the highest efficiency for transcriptional activation. We have demonstrated that EBV-positive tumor cells treated with CRISPR-Casilio activator system and sgRNA3 expressed the highest levels of BZLF1/Zta and its downstream lytic proteins. Therefore we have efficiently delineated the potential activator-binding regions in the *BZLF1* promoter, although the complexity of the system may hinder the effective delivery of the CRISPR-Casilio-based activator complex including 3 essential components to a patient.

In contrast to the CRISPR-Casilio-based activator, we selected the TALE-based translational activator system to develop an efficient lytic induction therapy because of its feasibility for *in vivo* therapeutic delivery. As the construction of sequence-specific TALE plasmids involve multiple cloning steps and validation processes, we designed our TALE construct TZ3 based on the sgRNA3-binding sequence identified from our successful test of the CRISPR-Casilio activator system to induce EBV reactivation. In our study, we also constructed additional TALE constructs (TZ1, TZ2, and T4) that targeted the corresponding binding sequences of the other sgRNAs to determine the concordance between the two systems. The consistent findings obtained using both systems demonstrated that the usefulness of the CRISPR-Casilio-based system in comprehensively screening for the effective transcriptional activator-binding sites. After proving the high efficiency and specificity of the TALE-based activator system in reactivating endogenous *BZLF1* expression in EBV-positive cancer cells, we further applied state-of-art nucleoside-modified mRNA and lipid nanoparticle (LNP) technologies to deliver the artificial *BZLF1*-specific transcriptional activator in our *in vivo* experiments. With this study, we have demonstrated a comprehensive strategy for the development of mRNA nanomedicine and synthetic transcriptional factors to activate potential therapeutic targets in human cancers.

To confirm the absence of off-target effects of the TZ3 transcriptional-activator, we conducted an RNA-sequencing analysis of TZ3-transfected and mTZ3-LNP-treated EBV-negative HK1 cells. The transcriptomics analysis revealed that no specific cellular target genes were activated by the *BZLF1*-specific transcriptional activator. No significant changes to the

transcriptome were observed during these experiments (Figure 2f and Figure 4a). Furthermore, we conducted ChIP-sequencing using a FLAG-specific antibody to identify the binding sequences of the *BZLF1*-specific transcriptional activator in the cellular and EBV genomes from EBV-positive SNU719 cells treated with mTZ3-LNP. Our new ChIP-sequencing experiments indicated that the predicted TZ3-binding sequence in the *BZLF1* promoter was highly enriched. Although ChIP-sequencing experiment also detected sequence enrichment in a region on human chromosome 16q11.2 (chr16:46394685-46395085), the only potential binding sequence is near the centromere of chromosome 16, far (>100 kb) from the predicated promoter or enhancer of adjacent genes. Consistent with the RNA-sequencing results, the findings thus support the absence of an off-target effect of the TZ3 synthetic transcriptional activator (Figure 5c).

Reviewer #3:

Overall comments:

In this manuscript, Wu *et al.* developed the CRISPR-Casilio and TALE activator to activate the expression of the *BZLF1* gene that can reactivate the lytic cycle and induce cytolytic effect in EBV-positive epithelial cancers. The nucleoside-modified mRNA encoding TALE lytic activator was delivered by LNP into cells and in vivo, and the therapeutic efficacy was demonstrated. Overall, the experimental designs were logical. The conclusions were partly supported by the observed experimental results and more descriptions for some experiments should be provided to help the reader understand the research. Several concerns as shown below need to be addressed before it's suitable for publication.

Comments:

1) In line 144, when testing the the CRISPR-Casilio system, the authors stated that “Treatment of stably transfected cells with doxycycline (Dox) resulted in transactivator induction and the abundant expression of Zta, Rta, and downstream lytic proteins (BGLF4 and VCA) in EBV-positive epithelial cancer cells”. However, in the data shown in Fig 1d, among 2 out of the 3 cell lines tested, the BGLF4 and VCA were barely expressed or not expressed at all based on western blotting. In Supplementary Fig 2a, based on qPCR, BGLF4 also was not expressed in C17 cells. Therefore, the statement was not supported by the observed results. For therapeutic purposes, can the authors comment on whether this a concern? Although these downstream lytic proteins were not induced at all in C17 cells, the inhibition of cell viability and colony formation seemed to be as effective as in other 2 cell lines. Was it due to that other lytic proteins were induced in C17 cells to achieve this effect and what other lytic proteins? Was the induction of downstream genes depending on the cell lines and will it lead to inconsistency of the lytic effects? Similar results were observed when testing TALE based activator in C666-1 cells in Fig 2b.

Our reply:

Thank you very much for the comment. We have revised the indicated sentence to clearly describe the results of our experiments using the inducible CRISPR-Casilio system. In the revised manuscript, we have stated that “The treatment of stably transfected SNU719 and C17 cells with

doxycycline (Dox) resulted in transactivator induction and the expression of Zta, Rta, and downstream lytic proteins (e.g., BGLF4, EA-D, VCA and gp350) in these EBV-positive epithelial cancer cells (Fig. 1d-f, Supplementary Fig. 2a)” and “In stably transfected C666-1 cells treated with Dox, no late lytic proteins (VCA and gp350) were detected, whereas immediate early (Zta, Rta) and early (BGLF4) lytic proteins were induced (Fig. 1d-f)”.

We used three EBV-positive epithelia cancer cell lines were used in our studies involving the inducible CRISPR-Casilio-based systems. Despite the effective induction of BZLF1 expression, we observed different efficiencies of Rta and early and late lytic gene induction between the three EBV-positive epithelial cancer cell lines (SNU719, C666-1 and C17). The induction of Rta, early and late lytic gene expression by BZLF1, following its induction by the artificial transcriptional activator, was influenced by the copy number and single nucleotide variants of the EBV genome in these cell lines. As mentioned in our reply to Reviewer 1, C666-1 harbors a nonsense mutation in an EBV lytic gene, *BNRF1*, resulting in a deficiency in late lytic gene expression and virion production (Hui, et al, 2012; Tso et al, 2013). C17 is a cell line derived from a metastatic NPC, contains only 2-3 copies of EBV genomes per cell. As shown in the Figure 1d, the Dox-induced activator induced both early (BGLF4) and late (VCAp18) lytic protein expression at 96 h post-treatment. In Supplementary Fig 2a, induction of EBV lytic gene transcripts, including *BGLF4*, were detected at 48-96 h post-Dox-treatment. Because C17 cells have a low EBV copy number, they require a longer period (relative to the other cell lines) to accumulate sufficient Zta and Rta proteins for the transcriptional activation of downstream early and late lytic genes. As mentioned in the reviewer’s comment, similar inhibition of cell viability and colony formation was observed in Dox-treated stably transfected C17 cells and two other cell lines (SNU719 and C666-1) following the expression of the CRISPR-Casilio-based *BZLF1* activator. Similar *in vitro* and *in vivo* inhibitory effects of the TALE-based transcriptional activator (TZ3) on the growth of these EBV-positive epithelial cell lines are also shown in Figure 5e and Figure 7. According to previous studies, the overexpression of EBV immediately early and early lytic proteins contributed to the killing of the NPC and gastric cancer cells. For example, Zta was shown to induce G2/M arrest and p53 phosphorylation in tumor cells. Studies have reported that Rta induces irreversible G1 arrest and cellular senescence in NPC cells. Other early viral proteins, such as EBV alkaline exonuclease BGLF5 can switch off most cellular gene functions and trigger the apoptotic machinery. As shown in our *in vitro* and *in vivo* studies, our synthetic *BZLF1*-targeted transcriptional activator efficiently suppressed the growth of tumors generated from the EBV-positive epithelial cancer cell lines, even those with a low copy number of EBV episomes or a deficiency in late lytic gene induction (i.e. abortive lytic cycle).

References:

- Hui KF, Ho DN, Tsang CM, Middeldorp JM, Tsao GS, Chiang AK. Activation of lytic cycle of Epstein-Barr virus by suberoylanilide hydroxamic acid leads to apoptosis and tumor growth suppression of nasopharyngeal carcinoma. *Int J Cancer*. 2012;131(8):1930-40.
- Mausier A, Saito S, Appella E, Anderson CW, Seaman WT, Kenney S. The Epstein-Barr virus immediate-early protein BZLF1 regulates p53 function through multiple mechanisms. *J Virol*. 2002; 76: 12503–12.

- Mauser A, Holley-Guthrie E, Simpson D, Kaufmann W, Kenney S. The Epstein-Barr virus immediate-early protein BZLF1 induces both a G(2) and a mitotic block. *J Virol.* 2002; 76: 10030–7.
- Feng WH, Westphal E, Mauser A, Raab-Traub N, Gulley ML, Busson P, Kenney SC. Use of adenovirus vectors expressing Epstein-Barr virus (EBV) immediate-early protein BZLF1 or BRLF1 to treat EBV-positive tumors. *J Virol.* 2002; 76: 10951–9.
- Wang L, Shan L, Lo KW, Yin J, Zhang Y, Sun R, Zhong J. Inhibition of nasopharyngeal carcinoma growth by RTA-expressing baculovirus vectors containing oriP. *J Gene Med* 2008; 10: 1124–33.
- Chen YJ, Tsai WH, Chen YL, Ko YC, Chou SP, Chen JY, Lin SF. Epstein-Barr Virus (EBV) Rta-mediated EBV and Kaposi's sarcoma-associated herpesvirus lytic reactivations in 293 cells. *PLoS One*, 2011; 6: e17809.
- Zuo J, Thomas W, van Leeuwen D, Middeldorp JM, Wiertz EJ, Rensing ME, Rowe M. The DNase of gammaherpesviruses impairs recognition by virus-specific CD8+ T cells through an additional host shutoff function. *J Virol.* 2008; 82: 2385–93.

2) In Supplementary Fig 2a, what is the role of triton X-100 detergent? Does it help or disrupt the LNP formation? From the results, it looked like it's required for the LNP formation. Maybe the authors should be described to help readers not in the LNP delivery field understand.

Our reply:

In Supplementary Figure 2a of our original manuscript (Supplementary Figure 3a in the revised version), we illustrated the process for determining the encapsulation efficiency of the formulated mRNA-LNPs. When mRNAs are encapsulated by the formulated LNP, it is expected that no mRNA could be detected on agarose gel and by quantitative analysis (marked as Encapsulated with LNP). Triton X-100 detergent was used to disrupt the assembled mRNA-LNP to release the mRNAs for detection, either control- and mTZ3 mRNAs. As shown in the original Supplementary Figure 2a, the lane marked “Encapsulated with LNP and + Triton X-100” contains mRNA bands that appeared in the agarose gel at a similar size and signal level as that in the lane marked “Input mRNA”. By comparing the amount of input mRNA with the amount of mRNA released from the Triton X-100-treated LNPs (marked as “Encapsulated with LNP and + Triton X-100”), the encapsulation efficiency of the assembled mRNA-LNP could be determined. According to the reviewer's suggestion, we have described this process in detail in the legend of Supplementary Figure 3a in our revised manuscript.

3) The order of supplementary figures should be arranged following the order of description in the text – Fig S3c was actually discussed after Fig S4 and S5. Also, in FigS3c, would the high uptake of LNP-encapsulated mRNA in the liver be a concern?

Our reply:

Thank you for the comments. In the revised manuscript, we have arranged the figures according to the order in which they are described. The original Supplementary Figure S3c has been moved to Supplementary Figure 7a in the revised version. As shown in Figures 2d-h, 5a and 5d-e, the TZ3 TALE transcriptional activator is specific for the *BZLF1* promoter in EBV genome, with no obvious cytotoxic effects in EBV-negative cells. The absence of an off-target effect of TZ3 transcriptional activator in the treated cells was demonstrated in our study (Figure 2f, 5a and 5d). Normal liver cells are EBV-negative, and neither EBV lytic proteins nor latent *EBERs* were detected in the liver tissues of NOD-SCID mice treated with mTZ3-LNPs. Supplementary Figure 11b illustrates the absence of BZLF1 expression in the liver tissues of mice treated with mRNA-LNP-based lytic induction therapy. Despite the high uptake of LNP-encapsulated mRNA by liver tissues, the mRNA and TALE activator would be degraded subsequently and should not be a concern. In our *in vivo* study, we observed no obvious effects on the liver tissues and liver function markers (ALT, AST) in NOD-SCID mice treated with multiple doses of mTZ3-LNP and/or GCV (Supplementary Figure 10a and 10c).

4) In Fig 5, what is the significance of EA-D and BMRF1? Why checking these proteins? It should be explained in the text.

Our reply:

EA-D is a viral early lytic protein encoded by *BMRF1*. Similar to BGLF4, it is a downstream target of the Zta and Rta transcription factors and a key component in EBV lytic genome replication. In Figure 5 of original manuscript, we showed the detection of EA-D protein expression to confirm that treatment with mTZ3-LNPs *in vivo* also successfully reactivated the early lytic protein in the tumor tissues, in addition to the immediate early lytic protein Zta. Furthermore, an anti-BGLF4 antibody for immunohistochemical staining is not available. Thus, we conducted the RNAscope in-situ hybridization to illustrate the *in vivo* effect of mTZ3-LNPs in terms of inducing the mRNA transcription of two EBV early lytic genes, *BMRF1* and *BGLF4*.

As shown in Figure 6a-d of our revised manuscript, the *in vivo* effects of mTZ3-LNPs on the induction of Zta/BZLF1 expression, as well as a panel of downstream early (EA-D/BMRF1, BGLF4) and late lytic (gp350/BLLF1) genes in the tumor tissues were determined using immunohistochemical staining and RNAscope in-situ hybridization. We have included and explained the objectives of these experiments in the Results section of our revised manuscript.

5) There are several typos and grammatical errors, the authors should carefully proofread. For example:

Line 98: “”board-spectrum

Line 113: activation “of” and proven the usefulness

Line 217: Fig 2e should be Fig 2f

Line 295: could not be effectively “induce” when

Our reply:

The revised manuscript has been proofread and edited by a professional editor. We have checked the final version of the manuscript and corrected the typos and errors.

Reviewer #4:

Overall comments:

In this manuscript, authors reported an mRNA therapeutic approach for targeted therapy of EBV-positive cancers. Authors designed an mRNA encoding the BZLF1-specific transcriptional activator (mTZ3) and prepared lipid nanoparticles (LNPs) to encapsulate the mTZ3 for systemic delivery (mTZ3-LNPs). Authors demonstrated that cellular treatment of mTZ3-LNPs activated EBV lytic gene expression and intravenous injection of mTZ3-LNPs induced selective anti-tumor effects in EBV-positive tumors via mRNA-based lytic induction. Overall, this manuscript is well written and showed solid results to support the conclusions. However, I would like to suggest that authors consider following comments to improve the manuscript before publication.

Comments:

1) In figure 5, authors only showed the expression of EBV lytic genes in the tumor tissue sections after mTZ3-LNP injection. If authors provide data on the cellular distribution of LNPs in the tumor tissues, it will help readers understand how mTZ3-LNPs induce the anti-tumor effects. And cellular expression of EBV lytic genes should be also studied by flow cytometry.

Our reply:

Thank you for the reviewer's comments. As shown in Figure 3b, the uptake of mTZ3-LNPs and dissociation of the components occur rapidly in the tumor cells. Although the half-life of the formulated LNPs in circulation is around 8 hours, the uptake of LNP and dissociation process are dynamic (Supplementary Figure 6b). Although we can detect the fluorescent labeled LNP in the tumors of mice at 3 h post-injection of the Dil C18 labeled LNPs, the dissociation process was ongoing. It would not be feasible for us to accurately determinate the distribution of LNPs in the FFPE sections from the tumor tissues at 12, 24, 48 h post- treatment. As described in the Discussion section of our revised manuscript, Zta proteins induced by the transient expression of the synthetic TZ3 transcriptional activator could initiate a positive feedback loop to active the multiple *BZLF1* promoters in the EBV-positive tumor cells, resulting in the accumulation of Zta even when the TZ3 transcriptional activators are absence in the later time points. As shown in the Figure 6a-d of our revised manuscripts, increase of the percentages of Zta-expressing cells were shown from 12 to 48 h post-treatment. In Figure 6c, the heterogenous expression pattern of *BZLF1* mRNA indicated that the status of mTZ3-LNP induced lytic cycle was changing over time. The RNA *in-situ* hybridization experiments demonstrated the initiation of *BZLF* transcriptions in the tumor cells even at 48 h post-treatment. The unique "kick and accumulation" mechanism may facilitate the efficient reactivation of EBV lytic genes by the mTZ3-LNPs. In these experiments, we further determined the percentages of tumor cells with the expression of multiple EBV lytic proteins (Zta,

EA-D and gp350) and with the transcription of lytic gene mRNAs (*BZLF1*, *BGLF4*, *BMRF1*, *BLLF1*) in tumor sections using imageJ software to analyze our immunohistochemical staining and RNA-*in situ* hybridization results. In a pilot study, we also evaluated the expression of Zta genes in the SNU719 tumors at 48 h post-mTZ3-LNP injection using flow cytometry analysis. Flow cytometry analysis revealed a relative higher percentage of Zta-positive cells (26%-28%) in the dissociated cells from the tumors after mTZ3-LNP treatment when compared with the IHC and RNA-*in situ* hybridization experiments (figure below). However, this result may be attributable to the problems in cell dissociation and the loss of fragile tumor cells during the tissue processing of treated tumor tissues. We believe that the quantitative analyses of IHC and RNA-*in situ* hybridization results are better to illustrate the effects of *in vivo* mTZ3-LNP treatment on the induction of EBV lytic gene expression in tumor xenograft models, as shown in Figures 6a-d in our revised manuscript.

2) To demonstrate the clinical translatability of mTZ3-LNPs, it is necessary to compare the therapeutic efficacy of mTZ3-LNPs (+GCV) with conventional treatments for EBV-positive cancers, particularly in *in vivo* anti-tumor studies.

Our reply:

In this study, we demonstrated the potent antitumor effect and safety of mTZ3-LNP treatment by applying it to EBV-positive cancer cell lines derived from primary tumors (SNU719, C666-1), local recurrence (Xeno-76) and distant metastasis (C17). While approximately 10% of all gastric cancers are EBV-positive, there are no specific treatment strategies and targeted therapies for this unique cancer subtype. As reported in previous studies, the primary NPC cell line C666-1 is

resistant to conventional chemotherapeutic agents (e.g. Cisplatin, Docetaxel, Fluorouracil) (Busson, et al, 1988). As noted, the xeno-76 PDX and C17 cell line were established respectively from the recurrent and metastatic tumors of NPC patients who had previously received multiple conventional treatments, including radiotherapy, chemoradiotherapy and polychemotherapy (Lin, et. 2018; Hsu et al, 2015; Busson et al, 1988). In other words, these recurrent or metastatic NPC models were derived from treatment-resistant tumors. Strikingly, these aggressive NPC xenograft models responded dramatically to mTZ3-LNP-based lytic induction therapy. As discussed in our recent review article (Wong, 2021), the poor prognosis of patients with recurrent and metastatic NPC and the various long-term adverse effects of treatment are critical problems facing the clinical management of NPC. Our preclinical findings suggest that our innovative mRNA-LNP-based treatment strategy could improve the clinical outcomes of patients with recurrent and metastatic NPC. We expect that future clinical studies will help to confirm the usefulness of this mRNA nanomedicine for the first-line treatment of such patients.

References:

Busson P, Ganem G, Flores P, Mugneret F, Clausse B, Caillou B, Braham K, Wakasugi H, Lipinski M, Tursz T. Establishment and characterization of three transplantable EBV-containing nasopharyngeal carcinomas. *Int J Cancer*. 1988 Oct 15;42(4):599-606.

Lin W, Yip YL, Jia L, Deng W, Zheng H, Dai W, Ko JMY, Lo KW, Chung GTY, Yip KY, Lee SD, Kwan JS, Zhang J, Liu T, Chan JY, Kwong DL, Lee VH, Nicholls JM, Busson P, Liu X, Chiang AKS, Hui KF, Kwok H, Cheung ST, Cheung YC, Chan CK, Li B, Cheung AL, Hau PM, Zhou Y, Tsang CM, Middeldorp J, Chen H, Lung ML, Tsao SW. Establishment and characterization of new tumor xenografts and cancer cell lines from EBV-positive nasopharyngeal carcinoma. *Nat Commun*. 2018 Nov 7;9(1):4663.

Hsu CL, Kuo YC, Huang Y, Huang YC, Lui KW, Chang KP, Lin TL, Fan HC, Lin AC, Hsieh CH, Lee LY, Wang HM, Li HP, Chang YS. Application of a patient-derived xenograft model in cytolytic viral activation therapy for nasopharyngeal carcinoma. *Oncotarget*. 2015 Oct 13;6(31):31323-34.

Wong KCW, Hui EP, Lo KW, Lam WKJ, Johnson D, Li L, Tao Q, Chan KCA, To KF, King AD, Ma BBY, Chan ATC. Nasopharyngeal carcinoma: an evolving paradigm. *Nat Rev Clin Oncol*. 2021 Nov;18(11):679-695.

3) Unlike the schematic in figure 7, co-treatment with antiviral ganciclovir did not augment the anti-tumor effect of mRNA-loaded LNPs in EBV-positive tumors. Why? Authors should explain this in Discussion section.

Our reply:

Thank you for the comments on the combined mTZ3-LNP and GCV treatment. We have revised the schematic in Figure 8 of the revised manuscript (Fig 7 of original version) to clearly describe the mechanism of action of the combined treatment. As shown in Figure 7 in the revised

manuscript, potent growth inhibitory effects were observed in the tumors treated with mTZ3-LNP alone or in combination with GCV, but no significant difference in tumor volume/tumor index was observed between the two treatment groups. The absence of an obvious bystander killing effect observed in the combined mTZ3-LNP+GCV treatment group is believed to be due to the high lytic induction efficiency and potent cytotoxicity of mTZ3-LNPs in EBV-positive tumor cells. We have explained our observations in the Discussion section as follows:

“Through *in vivo* study, we observed no significant difference in growth inhibition between tumors treated with mTZ3-LNP alone or in combination with GCV. The absence of an obvious bystander effect with combined treatment may be due to the high efficiency of lytic reactivation and the potent cytotoxic effect of mTZ3-LNP on EBV-positive tumors. Despite of the limited bystander killing effect observed, the GCV administration is an essential procedure for rapid and specific killing of the EBV-positive cells and inhibit the production of infective virions during EBV lytic induction treatment.”

4) As shows in supplementary figure 3, most of the injected LNPs accumulated in the liver. To support the safety of mTZ3-LNPs, authors should investigate whether intravenous injection of mTZ3-LNPs induces the expression of EBV lytic genes in the liver tissues.

Our reply:

According to the reviewer’s suggestion, we performed an immunohistochemical staining analysis to detect the BZLF1/Zta lytic protein in the liver tissues of mice treated with mTZ3/LNPs treatment (Supplementary Figure 10b). BZLF1/Zta protein was not detected in the liver tissues of mice injected with either mTZ3-LNP or control treatments. This finding confirmed the absence of EBV lytic reactivation in the liver tissues of the NOD-SCID mouse models implanted with EBV-positive tumor xenografts, thus supporting the safety of mTZ3-LNPs in the treatment of EBV-positive tumors.

Reviewer #5:

Overall comments:

The authors show an extensive work on the potential application of a lipid nanoparticle loaded with a mRNA in the EBV-associated cancers. The data confirmed the efficiency of the nanosystem *in vitro* and *in vivo*.

Comments:

1) The nanoparticles could be characterized by dynamic light scattering in order to confirm that the nanoparticles were formed as expected and to know the size distribution including the polydispersity index.

Our reply:

In response to the reviewer's comments, we conducted experiments to characterize the mTZ3-LNP nanoparticles. Using a Zetasizer Nano ZS90 system (Malvern Panalytical), we determined the average particle size of the mTZ3-LNP nanoparticles to be 124.1 ± 1.114 nm through dynamic light scattering (DLS) analysis. The polydispersity index (PDI) was 0.124 ± 0.032 . A representative size distribution is shown in Figure 3a. The data are representative of three independent experiments.

2) Does the nanoparticle pass through cell uptake process? Confocal microscopy of the cells incubated with fluorescent nanoparticles could prove it.

Our reply:

In Figure 3b, the cellular uptake of formulated lipid nanoparticle (LNP) encapsulated fluorescent labeled TZ3 mRNA (cy5-mTZ3-LNP) by the tumor cells after incubation was detected by confocal laser scanning microscopy (LSM 880, AxioObserver). LNP mediated endosomal/lysosome escape and cytoplasmic release of Cy5-labeled TZ3 mRNA in SNU719 cells were shown after a 3- and 6-h incubation with Cy5-mTZ3-LNP. The intracellular red signal derived from Cy5-TZ3 mRNA increased proportionally with the incubation time, suggesting a time-dependent cellular internalization.

3) Please include additional information about LSM measurements. Include equipment details and wavelength of excitation/emission.

Our reply:

We have included information about the use of confocal laser scanning microscopy (CLSM) to measure the intracellular uptake of LNPs by the tumor cells in the Methods section of our revised manuscript. The equipment used (LSM 880, AxioObserver). The fluorescence signals were measured in three channels: Cy5, excitation/emission wavelength, (ex/em) 633/697 nm; Dnd-26, ex/em, 488/524 nm; and hoechst, ex/em 405/460 nm.

4) The *in vivo* assays were performed for 2 weeks only. The authors could include justification for this choice.

Our reply:

In the *in vivo* experiments, we aimed to compare the growth inhibitory effects of the vehicle, mTZ3-LNP and GCV treatments on tumors in both the EBVaGC and NPC xenograft models. Thus, we harvested tumors from both the control and treatment groups for investigation and comparison once a tumor of the mice met the size limit (~ 1000 - 2000 mm³), according to the guideline of our animal ethics approval. As shown in Figure 7, the tumors grew rapidly in the mice, such that the tumors in the vehicle alone and vehicle+GCV control groups generally reached the

size limit at ~ 12-20 days post-treatment. We have added these details and our justification to the Methods section.

5) How long is the circulation time of the nanoparticles? The blood was analyzed by fluorescence?

Our reply:

We conducted an *in vivo* experiment in NOD-SCID mouse models to measure the circulation time of the formulated LNPs. After intravenously injecting the mice with the DilC18-labeled LNP encapsulated *TZ3* mRNA, we collected peripheral blood samples at 0, 8, 24 and 48 h for fluorescence signal determination. As shown in Supplementary Figure 8b, we determined the half-life of our formulated LNPs in circulation to be 8.34 h.

REVIEWERS' COMMENTS

Reviewer #1 (Remarks to the Author):

The authors have adequately addressed my initial concerns. I do not have additional comments to make.

Reviewer #2 (Remarks to the Author):

The authors have addressed my comments.

Reviewer #3 (Remarks to the Author):

All my concerns have been addressed.

Reviewer #4 (Remarks to the Author):

This manuscript has been significantly improved after this round of revision and I also have satisfied with authors' responses on my concerns. Thus, I would like to recommend it for publication in Nature Communications without further revision.